# Erythropoietin re-wires cognition-associated transcriptional networks

Manvendra Singh [1] ✉, Ying Zhao[1], Vinicius Daguano Gastaldi [1],
Sonja M. Wojcik[2], Yasmina Curto [1], Riki Kawaguchi[3], Ricardo M. Merino[4],
Laura Fernandez Garcia-Agudo[1], Holger Taschenberger [2], Nils Brose[2],
Daniel Geschwind [3], Klaus-Armin Nave [5] & Hannelore Ehrenreich [1] ✉

Recombinant human erythropoietin (rhEPO) has potent procognitive effects, likely hematopoiesis-independent, but underlying mechanisms and physiological role of brain-expressed EPO remained obscure. Here, we provide transcriptional hippocampal profiling of male mice treated with rhEPO. Based on ~108,000 single nuclei, we unmask multiple pyramidal lineages with their comprehensive molecular signatures. By temporal profiling and gene regulatory analysis, we build developmental trajectory of CA1 pyramidal neurons derived from multiple predecessor lineages and elucidate gene regulatory networks underlying their fate determination. With EPO as 'tool', we discover populations of newly differentiating pyramidal neurons, overpopulating to ~200% upon rhEPO with upregulation of genes crucial for neurodifferentiation, dendrite growth, synaptogenesis, memory formation, and cognition. Using a Cre-based approach to visually distinguish pre-existing from newly formed pyramidal neurons for patch-clamp recordings, we learn that rhEPO treatment differentially affects excitatory and inhibitory inputs. Our findings provide mechanistic insight into how EPO modulates neuronal functions and networks.

During strenuous motor-cognitive exercise, the brain as a whole and specific neuronal populations in particular, undergo physiological, "functional" hypoxia, with neurons experiencing relative deprivation of oxygen compared to their task-related requirements[1–6]. Functional hypoxia is an imperative developmental and physiological stimulus. It triggers key biological processes to compensate for consequences of oxygen deprivation and thereby adjusts to new requirements, among them erythropoiesis, promoting oxygen delivery for regaining homeostasis[7–10]. One of the major adaptive responses to hypoxia involves hypoxia-inducible factors (HIF), which bind and transcriptionally activate their responsive elements embedded in the

erythropoietin (EPO) gene[11–13]. In essence, the endogenous EPO system in the brain provides valuable fuel for individuals to renovate their synaptic and neuronal infrastructures upon facing metabolic challenges[2,14–20].

Previous studies, combining genomics, behavioral readouts, and functional assays, have illuminated the impact of brain EPO receptors and of EPO, injected or hypoxia-induced, as a driving force of "hardware upgrade" in particular of CA1 pyramidal neurons[16,17,21]. This "hardware upgrade"—even reflected by magnetic resonance imaging findings in mice and humans—comprises re-wired neuronal networks, enhanced dendritic spine density, and accelerated neuronal

[1]Clinical Neuroscience, Max Planck Institute for Multidisciplinary Sciences, City Campus, Göttingen, Germany. [2]Department of Molecular Neurobiology, Max Planck Institute for Multidisciplinary Sciences, City Campus, Göttingen, Germany. [3]Program in Neurogenetics, Department of Neurology, David Geffen School of Medicine, University of California Los Angeles, Los Angeles, CA, USA. [4]Max Planck Institute for Dynamics and Self-Organization and Campus Institute for Dynamics of Biological Networks, Georg-August-University, Göttingen, Germany. [5]Department of Neurogenetics, Max Planck Institute for Multidisciplinary Sciences, City Campus, Göttingen, Germany. ✉e-mail: manvendra.singh@mpinat.mpg.de; ehrenreich@mpinat.mpg.de; hannelore.ehrenreich@web.de

differentiation from pre-existing, nonproliferating precursors, resulting in an increment of CA1 pyramidal neurons of up to 20%, together with improved motor-cognitive performance[14,17,21–24]. It has, however, remained obscure whether the exposure to EPO results in widespread transcriptome alterations in hippocampal neuronal and non-neuronal lineages that, in turn, are causing the observed improvement in brain performance[25]. In addition, the molecular criteria defining the cellular response to EPO allegorize an enigma. It is also still unclear whether and to what extent EPO contributes to the maturation, migration and differentiation potential of early progenitor cells.

Single-nucleus RNA sequencing (snRNA-seq) quantifies the RNA repertoire in individual nuclei enabling the apprehension of both abundant and rare cell types from cryopreserved samples[26,27]. We hypothesized that snRNA-seq of mouse hippocampus would reveal the differential cell-type composition, gene expression patterns, and signaling pathways under the influence of EPO. Towards this goal, we performed unbiased snRNA-seq to procreate a single-nucleus transcriptomic landscape from EPO versus placebo (PL)-treated (control) hippocampal samples.

Here, we classify the major neuronal and non-neuronal cell types from the murine hippocampus using the computational efficiency of integrated snRNA-seq datasets from EPO and PL samples[28,29]. Based on our previous discoveries, we further investigated the changes that CA1 pyramidal neurons undergo after rhEPO treatment. Different from earlier work, we not only resolve the differentiation trajectory but with EPO as "tool" unexpectedly discover previously unseen pyramidal lineages with their transcriptional snapshots. We also extend our analysis to show the EPO-mediated transcriptome-wide response with the Gene Regulatory Networks (GRNs) enriching for neuronal differentiation, dendrite growth, synaptogenesis, memory formation, and cognition. EPO and PL transcriptomes resulted in being distinct from each other, suggesting the ability of EPO to substantially alter the transcriptome of pyramidal neurons, generally associated with migration and maturation. Our results indicate that two progenitors develop into mature CA1 pyramidal neurons. In the EPO samples, the differentiation of these newly formed neurons undergoes a complex series of transcriptional changes, causing greater abundance in the superficial niche in association with the higher differentiation potential. This is accompanied by the upregulation of migrating, and transsynaptic genes.

Finally, our single-cell electrophysiology experiments, based on the sophisticated genetic distinction between newly formed and preexisting CA1 pyramidal neurons, showed that rhEPO treatment differentially affected excitatory and inhibitory input to both, with newly formed neurons receiving more excitation and less inhibition under EPO treatment than pre-existing neurons.

Overall, our study highlights molecular and physiological underpinnings of how gene expression changes modulated by EPO are translated into substantial brain "hardware upgrade", including synaptic plasticity, and may help explain the consistently observed procognitive effects and notable performance adaptations in mice and humans.

## Results

### Remarkable diversity of pyramidal cell types in the mouse hippocampus

We commenced our study by examining the hippocampal transcriptional profiles of six EPO- and six PL-treated mice at the single-nuclei resolution (Fig. 1 and Supplementary Fig. 1). From the transcriptome of ~108,000 single nuclei, we first removed or adjusted the factors causing unwanted variation and then integrated those individual snRNA-seq datasets using "harmony"[28]. This approach cleaned the sequencing artifacts and normalized the batch effect quite robustly (Supplementary Fig. 1). Ultimately, using the combinations of established and most popular clustering algorithms[29–31], we assembled

transcriptionally similar nuclei, represented by 36 clusters (Fig. 2a, b), comprehended into ten major lineages and a neuroimmune cluster with distinct gene expression profiles: Oligodendrocytes, pyramidal neurons, interneurons, intermediate cells, Dentate gyrus neurons, astrocytes, microglia, endothelial cells, pericytes, and ependymal cells (Fig. 2c, Supplementary Fig. 2, and Supplementary Data 1).

Because this study aims to identify how EPO regulates the enrichment of pyramidal neurons, we restricted our further analysis to the pyramidal cluster. To achieve this, we similarly approached this data analysis as aforementioned, but with a subset of ~36,000 pyramidal nuclei (Supplementary Data 1). Integration of all the snRNA-seq pyramidal datasets shows the clusters based on their cell types and not by any sample, indicating control of batch effects (Supplementary Fig. 3). Our analysis refines the clusters at a more profound resolution than those represented in the existing databases. For instance, we identified 20 distinct cellular clusters, each defined by a unique set of marker genes, which we use as a reference to classify the diversity and heterogeneity within pyramidal lineages (Fig. 3a, b). We identify clusters straightforwardly corresponding to Dentate gyrus neurons, CA2, CA3, and even the more heterogeneous CA1 neuronal populations that needed further inspection (Fig. 3a, b and Supplementary Fig. 3). Upon closer examination, we resolve these subgroups further and classify them based on known transcriptome markers: deep and superficial (radial axis), dorsal and ventral (long axis), immature neurons, newly formed, migrating, serotonergic, and those with firing characteristics (Fig. 3c, d, Table 1, and Supplementary Fig. 3). Gene expression profiles used here for annotations are all experimentally validated key markers, reproduced in different laboratories and studies[32–44]. This allowed us to deliver a comprehensive gene expression atlas of the pyramidal layer from both EPO and PL hippocampus (Supplementary Data 2). We also provide a reliable marker list that distinguishes each lineage regardless of treatment (Supplementary Data 3). To sum up, our exploratory analysis provides a broader landscape of mouse hippocampus at the single-nuclei resolution that demonstrates more diverse and complex lineages than appreciated previously.

### EPO boosts the composition of newly formed pyramidal lineages

Using immunohistochemical quantification and a subtle genetic labeling approach, we had previously reported that mature pyramidal neurons increased in numbers by up to 20% in CA1 upon 3-week rhEPO treatment and that an increase in immature glutamatergic precursors was discernible at single-cell transcriptome level as early as 6 h after a single, intraperitoneal EPO injection[17,21]. However, the transcriptome employed there was from a few hundred single cells, thus, insufficient to capture the global composition of CA1 neurons. To determine the composition of CA1 neurons for each lineage at greater confidence, here we leveraged the transcriptome of ~36,000 pyramidal nuclei. We first segregated the PL and EPO nuclei from the "harmonized" datasets and then calculated the percent composition of each nuclei cluster. Consequently, by calculating the relative abundance of each cluster, we found that the composition was nearly overall consistent across both EPO and PL hippocampi (Fig. 3e, f, Supplementary Fig. 4, and Supplementary Data 4).

Nevertheless, in accord with our previous studies[17,21,45], we captured a slight but significant increment of dorsal mature CA1 neurons in the EPO samples. Remarkably, however, a cluster annotated as "Newly formed-migrating-superficial" (Nf.M.S) pyramidal neurons showed a dramatic enrichment of up to 200%, followed by the moderate enrichment of "Newly formed-migrating-superficial-Sox5" (Nf.M.S.Sox5) and "Newly formed-migrating-firing-serotonergic" (Nf.M.F.Ser) neurons (Fig. 3e, f, Supplementary Figs. 4 and 5a, and Supplementary Data 5). Taken together, our results reveal significant population shifts towards the early pyramidal lineages, thus indicating that EPO induces differentiation of the

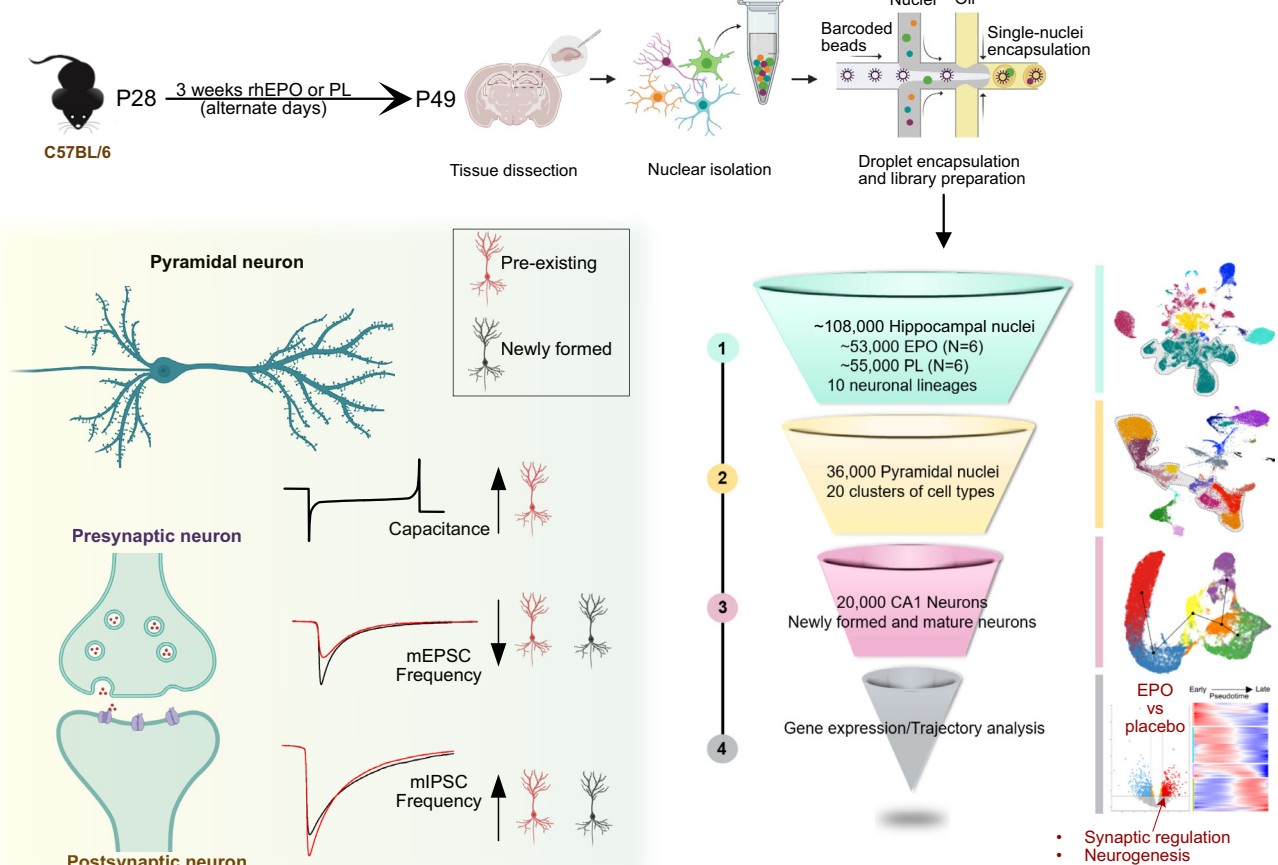

**Fig. 1 | Cartoon illustration of the present study design, workflow schematics, and overview of our findings.** Our study commences with the 3-week rhEPO ($N = 6$) and PL ($N = 6$) treatments in male C57BL/6 mice. After the last injection, we processed 12 samples, 6 each from EPO and PL mice, each a pool of 2 right hippocampi (i.e., a pool of 2 mice). These samples were then subjected to snRNA-seq, and the resulting data were strategically analyzed to investigate the molecular, pseudo-temporal, and empirical changes that occur in pyramidal lineages following EPO treatment. Finally, we performed a series of single-cell electrophysiology experiments to demonstrate that EPO affects excitatory and inhibitory input to newly formed and pre-existing pyramidal neurons in mouse hippocampi. Drop-seq design created with BioRender.com.

early stages of adult neurogenesis. Generally, because of sensitive differences in the frozen tissue dissociation, the percentages of these clusters may not resonate with their natural composition; nonetheless, we used a stringent method to calculate the significance level, and the shown cluster abundances are scaled[46–49]. Overall, our analysis demonstrates a physiological impact of brain-expressed EPO, reenacted by rhEPO treatment, through enriching the newly formed neurons in the CA1 region.

### Developing CA1 pyramidal neurons have multiple ontogenetic progenitors

Next, we asked if the newly formed neuronal clusters might be ontogenetic precursors to pre-existing CA1 neurons. To examine this, we considered two approaches[50]. First, we determined their pseudo-temporal dynamics and transcriptional states using *"Monocle"*[51]. Second, we tested the lineage decision trajectory using the *"Slingshot"* tool[52]. As revealed by both approaches, not just were the clusters placed in the same order, but also, they followed a similar path of differentiation trajectory. Both methods also uncovered that there are two progenitors leading to the fate of a single mature CA1 pyramidal neuron lineage (Fig. 4a, b and Supplementary Fig. 5b, c). To test if our trajectory analysis recapitulates the states of actual biological differentiation, we tested a set of gene expression dynamics over the pseudotime continuum (Fig. 4c and Supplementary Fig. 6a). These genes are markers and have established functional relevance in the

adult neurogenesis process: *Tbr1* and *Dcx* for newly formed or born neurons[39,44], *Sox5* for keeping the cells on progenitor state[40], *Cux1*, *Cux2, Reln, Ephrins*, and *Eph* receptors for migration[35,36,53,54], and *Zbtb20, Calb1, Ctip2,* and *Neurod6/Nex-1* for the various stages of maturation[32,55–57]. In summary, our analysis orders the nuclei according to their biological "pseudotime" progression, which might have derived from multiple transcriptionally distinct neuronal progenitors. Notably, upon further subgrouping the previous clusters, the common progenitor for mature CA1 pyramidal neurons turned out to be a distinct cluster (Supplementary Fig. 6b). Briefly, this fractionation strategy allowed us to discover high resolution of CA1 pyramidal branching where multiple progenitors converge to form a mature CA1 neuronal population. While it is tempting to resolve the molecular characteristics of each progenitor cluster, the scope of our study focuses on the role of EPO from hereon.

### EPO maneuvers the differentiation trajectory of pyramidal lineages

The above analysis permits us to ask whether the progression of progenitors to mature CA1 pyramidal neurons is identical for both EPO and PL samples. To this end, we assessed them separately on the differentiation axis (starting from *"Nf.M.S.Sox5"*, ending at *"mature CA1"*) inferred from *"Monocle"*. We detected that these neurons show an altered distribution, a significant shift in the nuclei density towards the earlier stages in EPO samples (Fig. 4d–f). This is likely driven by the cell

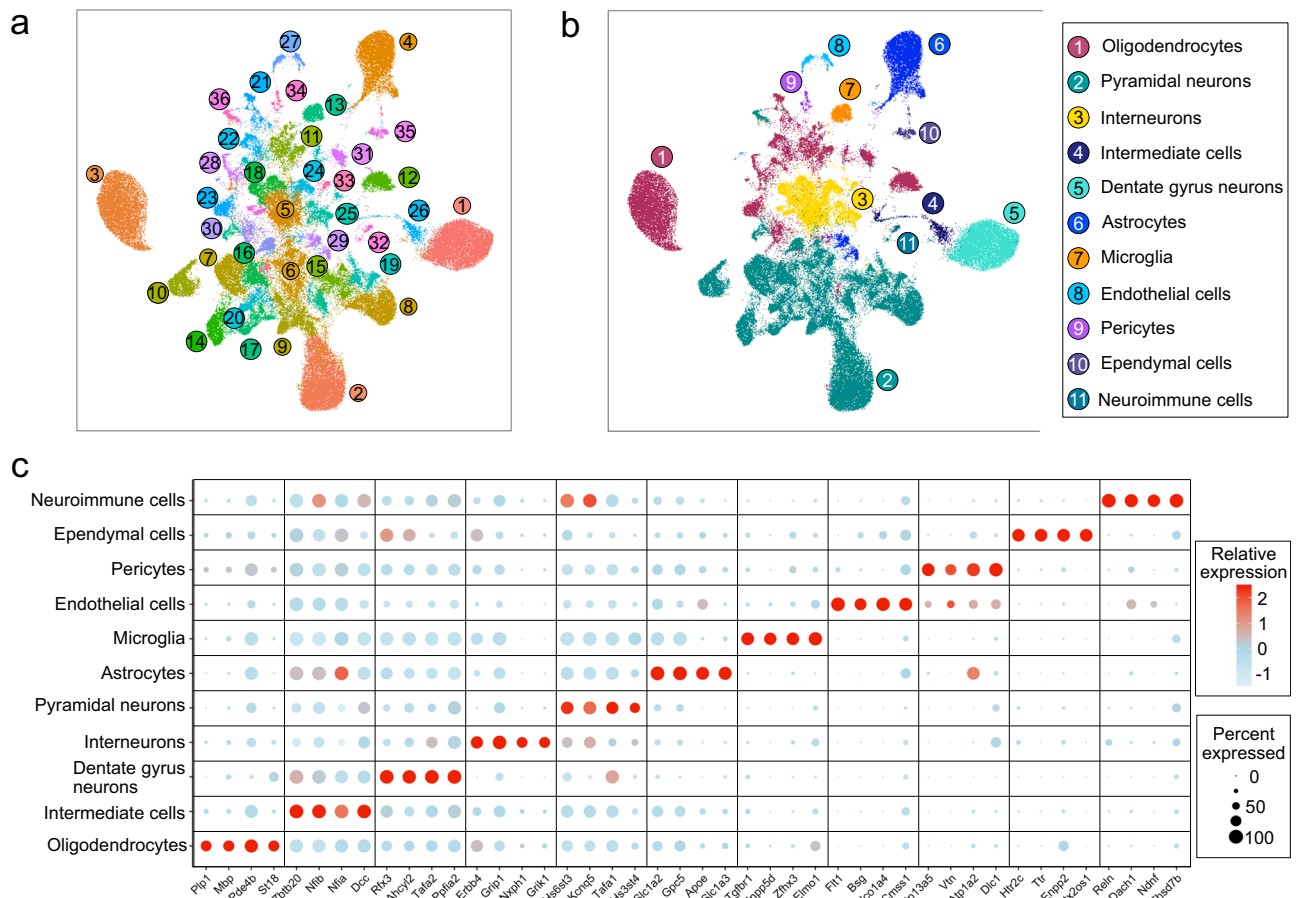

**Fig. 2 | Classification of neuronal and non-neuronal subpopulations from the mouse hippocampal nuclei landscape. a** Two-dimensional Uniform Manifold Approximation Plot (UMAP) resolving ~108,000 single nuclei, merged from each of 12 adult hippocampal samples treated with either EPO ($N = 6$) or PL ($N = 6$) into 36 different clusters. Colors indicate an unbiased classification of these nuclei via graph-based clustering, where each dot represents a single nucleus (see Supplementary Data 1 and Supplementary Fig. 1 for integrating EPO and PL samples). **b** The above 36 clusters on UMAP are consolidated into 11 major cell types based on

distinct expression patterns of known marker genes (see Supplementary Data 2 and Supplementary Fig. 2 for the clusters corresponding to each cell type). **c** DotPlot illustrates the intensity and abundance of mouse gene expression between the hippocampal lineage shown above. Colors represent an average Log2 expression level scaled to the number of unique molecular identification (UMI) values in single nuclei. The color scale ranges from light blue to red, corresponding to lower and to higher expression. Dot size is proportional to the percent of cells expressing that gene. Source data are provided on a repository[112] and as a Source Data file.

states of newly formed neurons. We found that Nf.M.S exist in a unidirectional branch, and according to previous observations, they showed a remarkable abundance of EPO nuclei. In the meantime, the Nf.M.F.Ser exhibited two branches, suggesting their existence in a dual cell state on the indistinct pseudotime. Along this axis, we discovered a significant enrichment of EPO samples in one of the two branches, expanding the arena where EPO might be influencing the lineage tree of CA1 neurons (Fig. 4e, f).

These results are consistent with the possibility that EPO fuels the differentiation potential of precursors, and therefore, we find a greater number of newly formed neurons. It still remains a mystery why the abundance of "mature CA1" neurons is relatively moderate upon rhEPO treatment (20%), compared to an up to 200% increase in a defined precursor demonstrated above. As we did not observe altered apoptotic events in our previous experiments[17], we discount the idea that newly formed neurons are simply prone to being excluded from the developmental process. One explanation could be that the impact of EPO on the newly formed neurons is encountered later in the development by innate processes such as homeostasis. We indeed detected relative affluence of a lineage in the middle of pseudotime trajectory in PL, suggesting a buffering process in EPO samples (Fig. 4e, f). Although the pseudotime of all lineages except mature CA1 cluster was relatively lower in EPO samples (Fig. 4g), this shift was noted most in the

"*superficial mature CA1*" neurons, the immediate predecessor of mature CA1 neurons on the pseudotime trajectory.

While we do not claim to fully decipher the mechanism of how the homeostasis is counteracting in EPO samples, we note the relative upregulation of *Tbr1*, and *Sox5* in "*superficial mature CA1*" neurons in PL samples, suggesting a second wave of neuronal differentiation (Fig. 5a). This coincides with the nuclei abundance along the middle of the trajectory in PL without their noticeable difference in the overall composition in the respective clusters. Altogether, we present a working model where rhEPO treatment forms a plethora of new neurons and then the pre-mature neurons are diminished so that the overall content of mature neurons remains affordable for the hippocampus.

## EPO modulates gene expression related to neurogenesis or synapses

The present data indicate that the composition and trajectory of pyramidal lineages modify under the influence of EPO. Might EPO be regulating the expression of host factors that are vital for neuronal, dendritic, and synapse development? To see a global picture of gene expression changes between EPO and PL hippocampi, we analyzed the snRNA-seq by apprehending them in pseudobulk format[58]. Briefly, we split the EPO and PL groups, then calculated average expression of

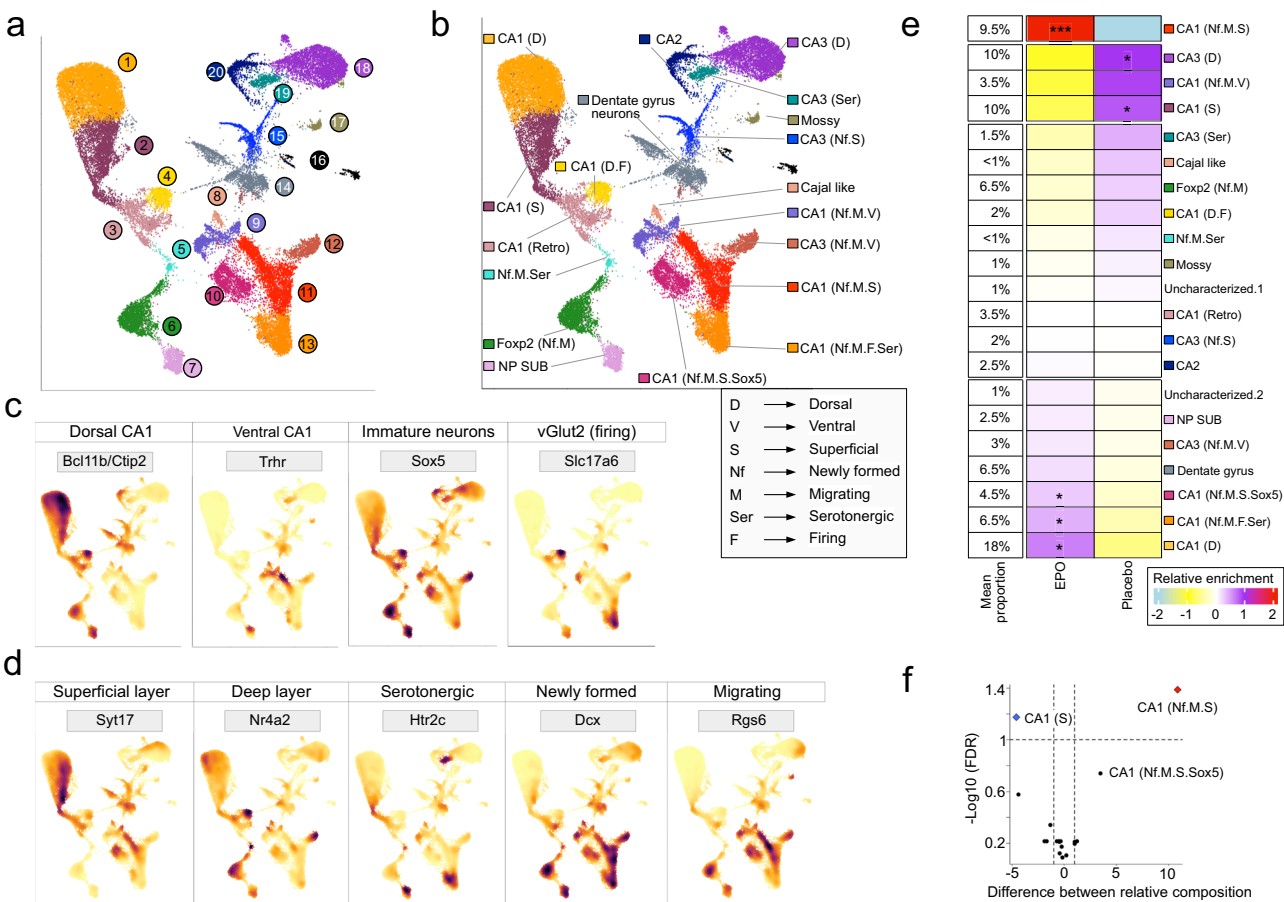

**Fig. 3 | Composition and classification of pyramidal cell types in the hippocampus from EPO and PL samples. a** Two-dimensional UMAP reveals ~36,000 single nuclei reanalyzed, classified as a pyramidal lineage in Fig. 2b. Using 2000 most variable genes, the top 30 principal components, our graph-based clustering resolves the 20 distinct cell populations shown in manually assigned color scales. Each dot represents a single nucleus. **b** Each cluster is shown in (**a**) is manually annotated based on the most discriminatory gene expression marking CA1, CA2, CA3, Dentate gyrus, and near-project subiculum (NP SUB) neurons. These clusters are further annotated based on marker genes for dorsal, ventral, superficial, and deep regions, or the markers of newly formed, migrating, serotonergic neurons shown inside the gray box. Clusters that expressed a detectible level of vGlut2 were classified as "firing" neurons. **c** Feature plots based on UMAP from (**b**) visualizing the expression of selected lineage-specific markers, e.g., *Bcl11b/Ctip2* (mature-dorsal CA1), *Trhr* (ventral CA1), *Sox5* (immature/progenitor neurons), and *vGlut2/ Slc17a6* (neurons with firing potential). **d** Feature plots based on UMAP from (**b**)

visualizing the expression of genes marking superficial and deep layers (*Syt17* and *Nr4a2*). The serotonergic, newly formed, and migrating neurons are labeled by *Htr2c, Dcx*, and *Rgs6*, respectively. **e** Heatmap illustrating the relative abundance of each pyramidal cell type shown in (**b**) in EPO and PL samples. The relative abundance was calculated by determining the observed fraction of each cell type compared to the expected fraction and using a two-sided Fisher exact test to identify cell types that were significantly enriched in EPO samples. Stars denote the *P* values, adjusted using the Bonferroni correction (*P* < 0.05= \**P* < 0.01= \*\**P* < 2.2e$^{-16}$ = \*\*\*). Note: the changes in neuronal composition are scaled values. **f** The linear model test over the relative composition of the lineages using the voom[113] method in the limma[49] R package. Volcano plot illustrating the average difference and false discovery rate (FDR) of each lineage shown in (**b**) between EPO and PL samples. FDR, here, is calculated by the Benjamini–Hochberg method. Source data are provided on a repository[112] and as a Source Data file.

## Table 1 | Abbreviation of distinct pyramidal sub-lineages

| CA (1|2|3) | cornu ammonis (1|2|3) |
|---|---|
| CA (1|3) (D) | CA1 or CA3 on the dorsal axis |
| CA1 (S) | CA1 in the superficial layer |
| CA1 (D.F.) | CA1 in the deep layer with firing characteristics |
| Nf.M | Newly formed and migratory |
| Nf.M.Ser. | Newly formed, migratory, and serotonergic |
| NP SUB | Near-projecting in SUBiculum area |
| Nf.M.V. | Newly formed and migratory on the ventral axis |
| Nf.M.S | Newly formed and migratory in the superficial layer |
| Nf.M.F.Ser | Newly formed, migratory, firing, and serotonergic |
| Nf.S | Newly formed in the superficial layer |

The abbreviation of pyramidal lineages was classified using snRNA-seq data analysis based on their predicted location and potential functional characteristics.

each gene across the nuclei clusters, and finally followed the edgeR pipeline to compute differentially expressed genes (DEG). To test whether both groups have a distinct transcriptomic profile, we compared their pseudobulk profiles on all pyramidal lineages pooled together. We observe the interindividual heterogeneous composition of neuronal subtypes, thus providing an additional layer to the mosaicism of pyramidal layers. Despite heterogeneity, hierarchical clustering followed by 1000 bootstraps of relative gene expression levels revealed that the EPO and PL transcriptomes are significantly divergent (Fig. 5b). We also noticed one EPO sample clustering with one PL sample, which we suspect is owing to the highly similar cell-type composition between them (Fig. 5b and Supplementary Fig. 7a— compare here eA1 and pB10). To generate a robust list of DEG, we ended up with five samples each after removing the one outlier per condition in our further analyses (see "Methods").

To investigate the genes and pathways most altered upon EPO, we performed DEG analysis in each of the six cell types shown on the

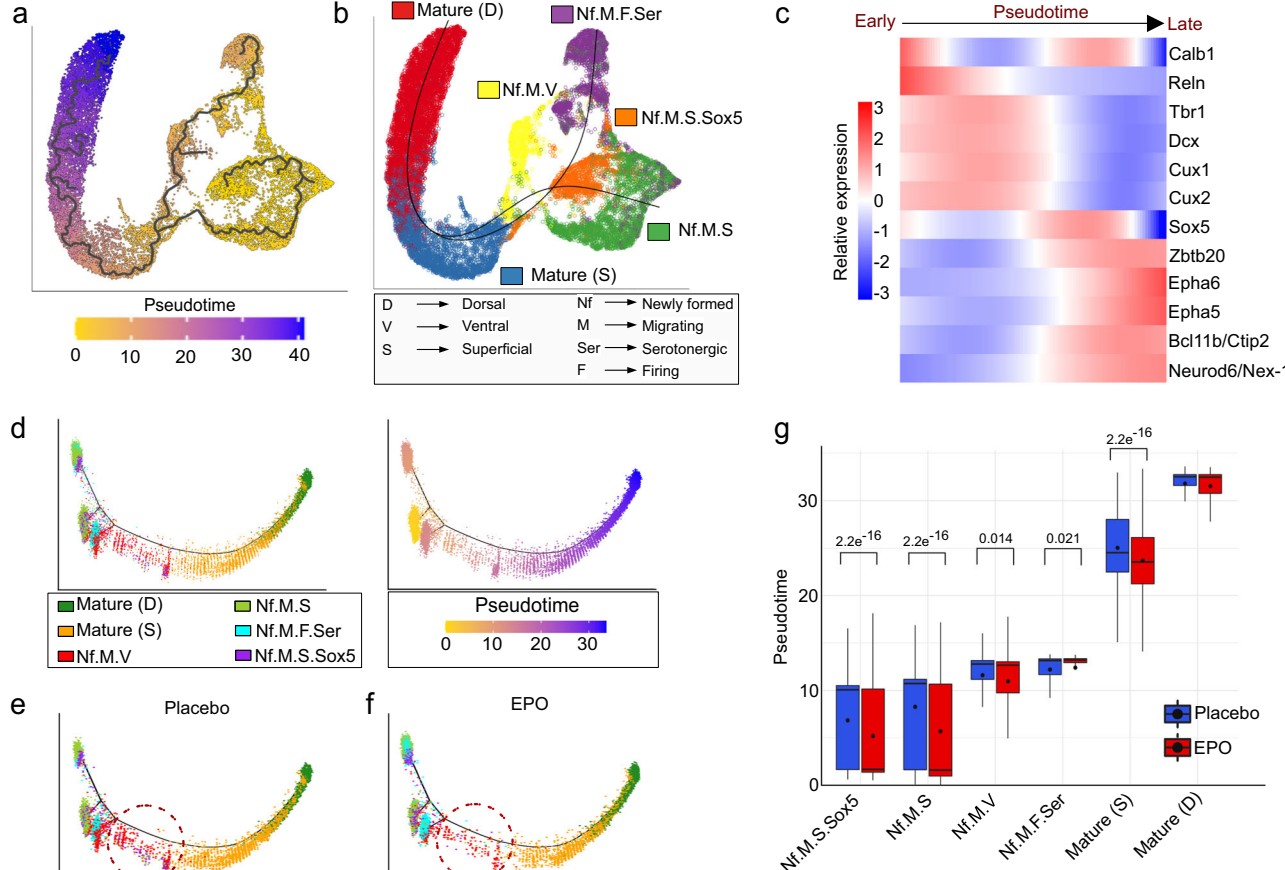

**Fig. 4 | EPO modulates the overpopulation of newly formed-migrating-superficial pyramidal neurons. a** Monocle2 single-nuclei trajectory analysis and nuclei ordering along an artificial temporal continuum using the differentially expressed genes between the pyramidal lineages shown in (**b**). The transcriptome from every single nucleus represents a pseudotime point along an artificial time vector that denotes the progression of newly formed to mature CA1 neurons. **b** Slingshot trajectory of analyzed pyramidal lineages that defines the branching out of newly formed neurons towards the mature CA1 neurons. Note that the artificial time point progression inferred from both Monocle2 and Slingshot agrees with the biological time points (also see panel **c**). **c** Heatmap showing the kinetics of genes changing gradually over the trajectory of newly formed to mature CA1 neurons. Genes (row) and nuclei (column) are ordered according to the pseudotime progression. Genes projected in early stages are associated with neuronal differentiation (*Tbr1, Dcx, Calb1*), neuronal migration (*Reln*), dendritic formation (*Cux1, Cux2*), and *Sox5* being the progenitor marker. The late stage is determined by

mature neuron and synapse formation genes *Epha5, Epha6, Bcl11b, Zbtb20,* and *Neurod6/NEX1*. **d** Pseudotime trajectory of the above pyramidal lineages shown on multifurcation tree obtained by default DDRTree parameters of Monocle2, colored from low (gold) to high (purple) pseudotime (right panel). Pyramidal lineage cell types clustered using the top 2000 differentially expressed genes and projected into a two-dimensional space (left). **e** For clarity, we show the above trajectory in two facets. Plot denotes PL. **f** Same as (**e**), except this plot denotes EPO samples. **g** Boxplot demonstrates the distribution of pseudotime values of the above lineages between EPO (*n* = 6) and PL (*n* = 6) samples. The colored boxes represent the interquartile range (IQR) divided by the median, and whiskers extend from minima to maxima of data by 1.5-fold of IQR beyond the box. EPO and PL samples were compared using the Wilcoxon Rank-Sum test, and Bonferroni adjusted *P* values are indicated for significant pairs only. Source data are provided on a repository[112] and as a Source Data file.

trajectory separately. We identified a varying number of significant (adjusted *P* < 0.05) DEG across the neuronal lineages (Fig. 5c and Supplementary Fig. 7b). We found that the number of DEGs is roughly inversely proportional to the pseudotime of each cluster, indicating the more significant impact of EPO at the transcriptional level in earlier rather than later stages. Also, even though several DEG are specific to a particular lineage, the pattern of their expression changes is primarily conserved across clusters (Fig. 5c, d). Among the conserved set of DEG in newly formed neurons, we find classic gene markers of neurogenesis and synaptic signaling, such as *Cux1, Cux2,* and *Homer1* etc. (Fig. 5c), pointing to global control of EPO over neuronal and synapse development.

To investigate the gene expression changes in detail, we focused on the Nf.M.S cluster that was most affected in EPO samples (Figs. 3e and 5c and Supplementary Fig. 7b). Because the general directionality and intensity of DEG were similar in all early lineages (Fig. 5d), the

impact of EPO in this cluster might reflect a general EPO effect on gene expression along the trajectories (Fig. 5c, d).

We found that 1043 genes (508 up, 535 down in EPO relative to PL) are significantly altered (adjusted *P* < 0.05) (Fig. 6a and Supplementary Data 6). The top candidates among those significantly upregulated genes are the vital genes for prescribing neuronal development, trans-synaptic signaling, and establishing proper cognition, learning and memory, such as *Arc, Homer1, Vgf, Egr1, Cux1, Egr3, Bdnf,* and *Syt4*[59–65] (Fig. 6a and Supplementary Fig. 8). *Cux1* and *Homer1* are not only upregulated in the EPO samples but are also among the most highly expressed genes in the "Newly formed-superficial-migrating" cluster (Fig. 6b and Supplementary Fig. 7c), suggesting that normal functions of this cell type such as dendritic development and synapse regulation are also enhanced upon EPO.

A compelling question is whether the same newly formed lineages in EPO and PL samples have different Gene Regulatory Networks

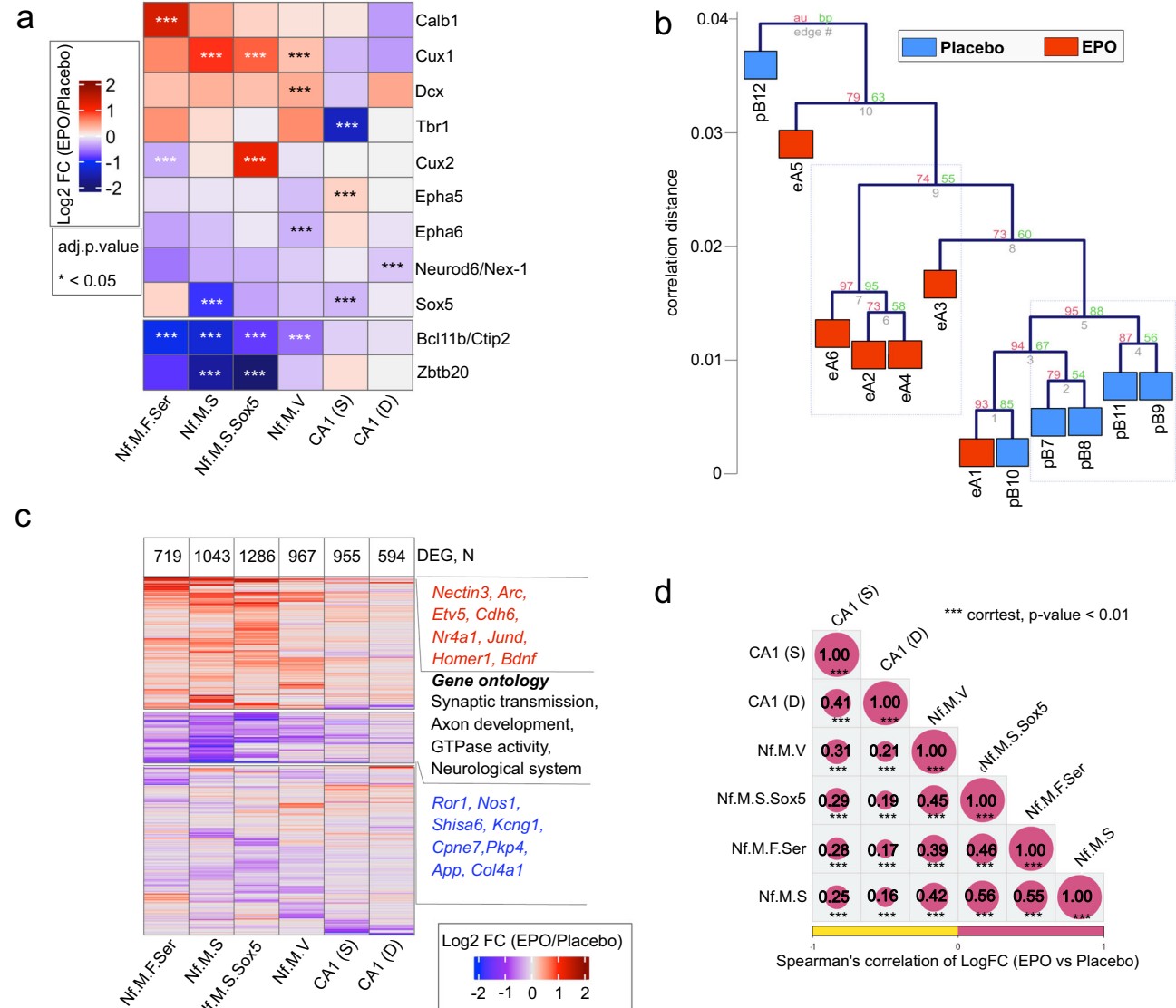

**Fig. 5 | EPO and PL samples reflect distinct transcriptomes. a** Heatmap representing the differential expression of selected genes (for clarity, see Fig. 4c) between EPO and PL samples in the individual CA1 pyramidal lineages shown in Fig. 3b. Data were presented as log2 fold change, $n = 5$, for each condition, stars denote the significant $P$ value < 0.01, adjusted using GLMLRTest from edgeR package and stars are indicated for significant $P$ values only. Exact $P$ values are provided in Supplementary Data 6 and in the Source data file. **b** Hierarchical clustering (Spearman rank correlation, average linkage) and bootstrapping (1000 replicates) of the transcriptomes of individual EPO (A01-A06) and PL (B07-B12) samples. Note that the clustering of A01 and B10 coincides with a similar proportion of lineages in those two samples (compare Supplementary Fig. 7a). **c** Heatmap representing the differential expression of genes between EPO and PL samples in the individual lineages shown in Fig. 4b. Total $n = 5$, for each condition, only those genes are presented that have significant $P$ value < 0.01, adjusted using GLMLRTest from edgeR package. The number of detected differentially expressed genes in each lineage is shown on the right side of the heatmap. **d** Correlation matrix displaying the pairwise comparison of differentially expressed genes between the comparisons shown in (**c**). The size of the circles is directly proportional to Spearman correlation strength (see the encircled values). Stars denote the significance levels of correlation at a $P$ value scale ***<0.05. $P$ values are obtained after "holm" adjustment for multiple tests of the pairwise Spearman correlation, tested using the corr.test function in R. Source data are provided on a repository[112] and as a Source Data file.

(GRNs). To uncover the GRNs in newly formed lineages, we executed SCENIC[66] in an unsupervised manner. The regulons in single nuclei were then grouped by their average scores per lineage separately for EPO and PL samples. Our analysis reveals that a large group of regulons were shared between the EPO and PL samples, with a couple of exceptions (Fig. 6c and Supplementary Data 7). Particularly, the regulons led by *Homer1* and *Nr4a1* were more specific to EPO samples. Both regulons shared the set of genes that are also known as immediate early genes (IEGs) and are involved in the recovery from neuropsychiatric disorders[67–75]. Thus, it further elucidates the potential of EPO to re-wire the GRNs associated with synaptic plasticity (Fig. 6c).

Overall, these distinct regulon activities might underlie clinical benefits observed upon EPO treatment[76–81].

## EPO changes the excitation/inhibition balance of CA1 pyramidal neurons

We now asked whether pyramidal neurons, either pre-existing or newly formed, would show distinct electrophysiological properties formed under the influence of rhEPO treatment. To distinguish these neurons for single-cell patch-clamp recordings, we used mice with tamoxifen-inducible reporter gene expression (NexCreERT2::R26R-tdT) (Fig. 7a). Tamoxifen administration before starting EPO/PL

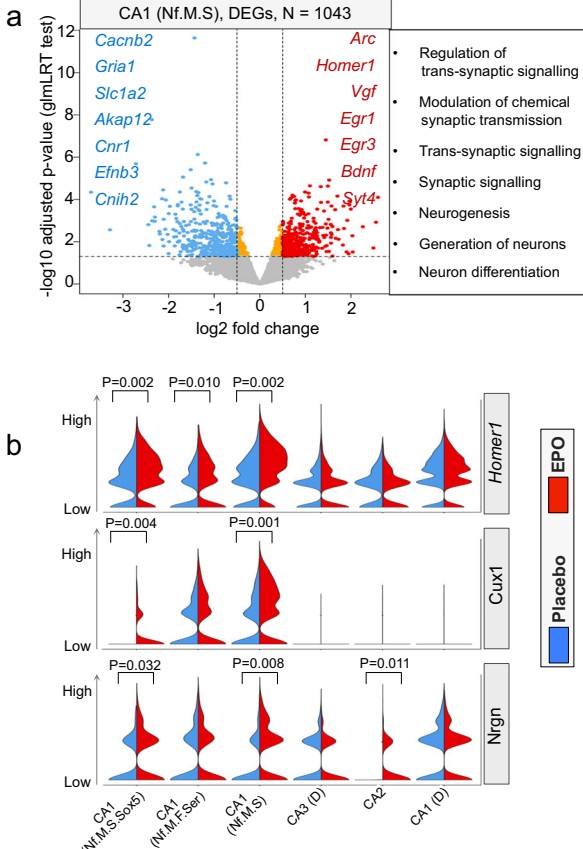

**Fig. 6 | EPO affects the trajectory and expression of genes involved in synaptogenesis and synapse function. a** Volcano plot showing genes that are differentially expressed between EPO and PL samples in newly formed and mature CA1 neurons. The horizontal dashed line indicates −Log10P = 2 (*P* value, adjusted using GLMLRTest). Boxed text beside the volcano plot corresponds to gene ontologies in which genes that are differentially expressed between EPO and PL cells are enriched. Exact P values are provided in Supplementary Data 6 and in the Source data file. **b** Violin plots showing the expression distribution of *Homer1*, *Cux1*, and *Nrgn* genes between PL (*n* = 5) and EPO (*n* = 5) samples in a pairwise fashion. *P* values are the results of GLMLRTest from edgeR package (see Supplementary Data 6),

Adjusted *P* values are obtained by Benjamini−Hochberg (BH) correction. **c** Barcoding the newly formed progenitor lineages in EPO and PL samples with regulons (gene sets that are inferred as GRNs). Heatmap shows the binary activity matrices obtained after applying the SCENIC tool. The activity status of the regulon in a particular lineage is presented as either active (black) or not active (white). Every row is a regulon where the master regulators are represented by their gene names. The number of genes enlisted in a respective regulon is in brackets. The full list of genes is provided in Supplementary Data 7. Source data are provided on a repository[112] and as a Source Data file.

treatment on P28 allows us to label essentially all mature pyramidal neurons previously present[21]. All pyramidal neurons differentiating and maturing after termination of the tamoxifen-induced Cre recombination lack tdTomato (Fig. 7a). Pyramidal neurons in the CA1 region of the hippocampus were analyzed by whole-cell patch-clamp recordings at P55, since we had previously found a considerable number of newly differentiated (tdTomato−/Ctip+) neurons in CA1, with about 20% more neurons upon EPO treatment, but no evidence by EdU incorporation of proliferating precursors, revealing adult "neurogenesis" independent of DNA synthesis[17,21].

Under EPO treatment, pre-existing neurons showed larger cell capacitance and lower input resistance when compared to newborn neurons (Fig. 7b, c), consistent with an increase in somatodendritic cell surface area, and likely reflecting the increase in dendritic spine formation found previously[21]. Although the effect on cell capacitance was statistically not significant with multiple comparison correction, input resistance, which is inversely proportional to cell surface area, and therefore cell size and complexity, showed significant effects for both EPO treatment (two-way ANOVA: PL vs EPO *P* = 0.0033) and neuron age (two-way ANOVA: old vs new *P* = 0.0011). Resting membrane potential (Fig. 7d), action potential (AP) threshold (Fig. 7e) and AP amplitude

(Fig. 7f) were not affected by EPO treatment or neuron age. However, EPO treatment did differentially affect the balance of excitatory and inhibitory input that pre-existing and newly formed pyramidal neurons receive (Fig. 7g–n). Neuron age significantly affected the amplitude of miniature excitatory postsynaptic currents (mEPSCs), with newly formed neurons receiving larger mEPSCs (two-way ANOVA: old vs new *P* = 0.0091). At the same time, EPO treatment significantly decreased mEPSC frequency for both old and new neurons (two-way ANOVA: PL vs EPO *P* = 0.0171). Decay time constants of mEPSCs were not significantly changed (Fig. 7j). Even more prominent differences were observed with respect to miniature inhibitory postsynaptic currents (mIPSCs) (Fig. 7k–n). For mIPSC amplitudes, we noted a significant interaction between treatment and neuron age (two-way ANOVA interaction term *P* = 0.005). Newly formed neurons received significantly smaller mIPSCs under EPO treatment than under PL (Fig. 7l). At the same time, the frequency of mIPSCs was significantly affected by both neuron age (two-way ANOVA: old vs new *P* = 0.0325) and treatment (two-way ANOVA: PL vs EPO *P* = 0.0008). Although EPO treatment increased mIPSC frequency for both pre-existing and newly formed neurons, the increase in mIPSC frequency was significantly greater for pre-existing neurons

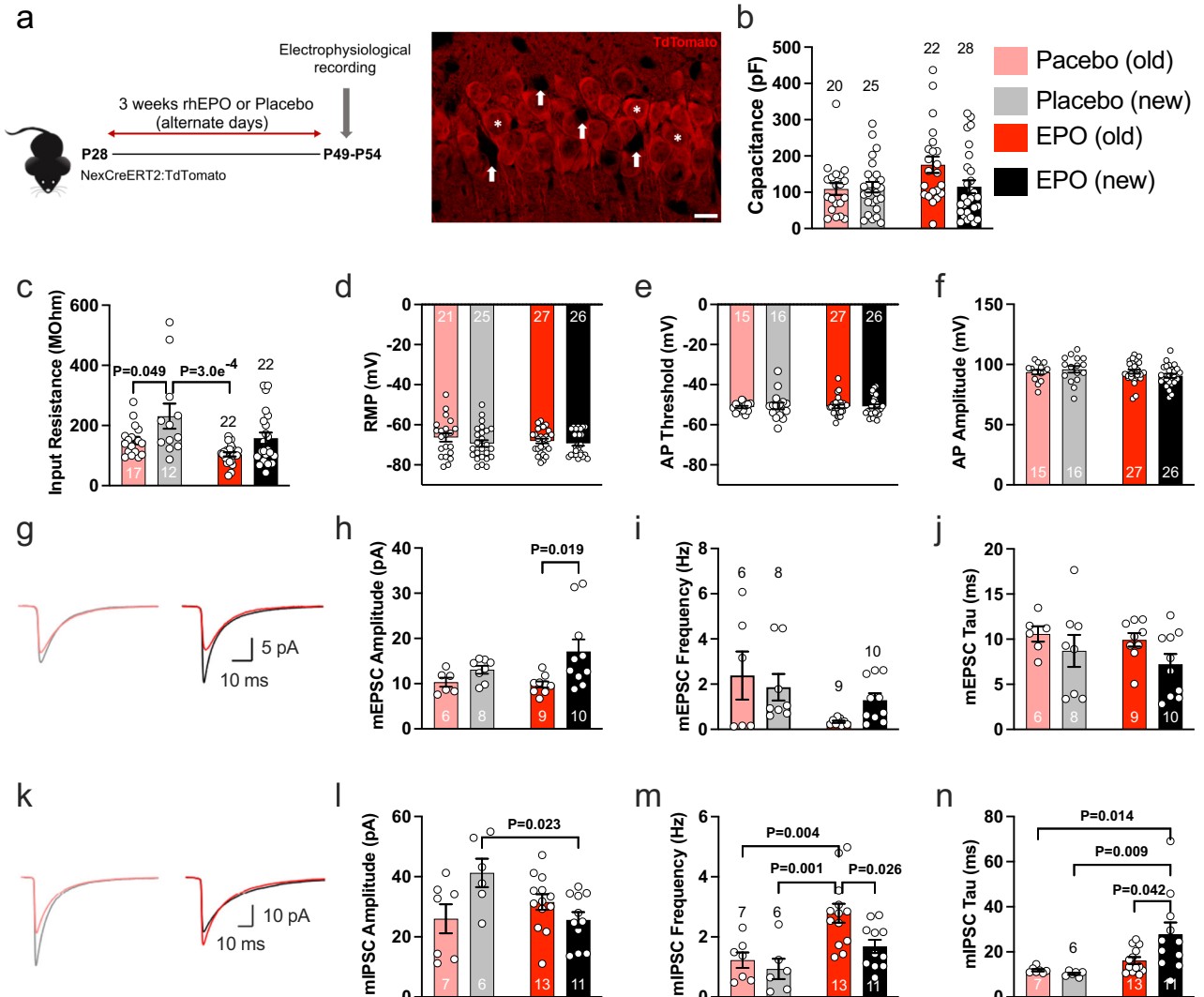

**Fig. 7 | EPO treatment differentially affects excitatory and inhibitory input to pre-existing and newly formed neurons. a** Schematic showing EPO/PL treatment regimen and immunohistochemical presentation of newly formed (arrows) and pre-existing (stars; tdTomato-expressing) CA1 neurons as used for the identification in patch-clamp recordings (scale bar: 10 μm). The identity of visually identified CA1 pyramidal cells was confirmed based on their passive membrane properties and their discharge behavior in response to depolarizing current steps.
**b** Comparison of capacitance for newly formed and pre-existing CA1 neurons from PL (PL)- and EPO-treated mice. Bar graphs show pre-existing (old) neurons in light red under PL treatment and in dark red under EPO treatment, and newly formed (new) neurons in gray (PL) and black (EPO) for (**b**–**n**). **c** Comparison of input resistance, which is inversely proportional to cell surface area. **d** Resting membrane potential (RMP) was not affected by EPO treatment or neuron age. **e** AP threshold was not affected by EPO treatment or neuron age. **f** AP amplitude was not affected

by EPO treatment or neuron age. **g** Averaged traces of mEPSCs from PL-treated (left) and EPO-treated (right) mice. **h** New neurons receive excitatory inputs generating larger mEPSC amplitudes under EPO than old neurons. **i** Comparison of mEPSC frequencies. **j** Comparison of mEPSC decay times. **k** Averaged traces of mIPSCs from PL-treated (left) and EPO-treated (right) mice. **l** New neurons receive inhibitory inputs generating smaller mIPSC amplitudes under EPO treatment than under PL. **m** New neurons receive lesser increase in mIPSC frequency on EPO than old neurons. **n** mIPSC decay time constants increase under EPO treatment. Bar graphs show mean values with SEM as error bars, the number of cells analyzed is listed within each bar. Statistical analysis was two-way ANOVA (PL vs EPO and old vs new) with Tukey's method for multiple comparison correction. *P* values of the two-way ANOVA are reported in the text, and *P* values from Tukey's method are represented in the figure panels. Source data are provided as a Source Data file.

(Fig. 7m). EPO treatment also resulted in a significant increase in mIPSC decay time constants for both pre-existing and newly formed neurons (two-way ANOVA PL vs EPO: $P = 0.004$) (Fig. 7n). Overall, newly formed neurons received more excitatory input (Fig. 7h–i) and at the same time less inhibitory input with a reduction in mIPSC amplitude (Fig. 7l) and a smaller increase in mIPSC frequency (Fig. 7m) under EPO treatment than pre-existing neurons. Thus, EPO treatment has a striking differential effect on the balance of excitation and inhibition received by pre-existing and newly formed neurons, and is therefore expected to determine how newly formed neurons integrate into the existing

neuronal networks and may potentially affect hippocampal information processing.

## Discussion

The present study originally aimed to shed light on how EPO regulates the enrichment of pyramidal neurons. To answer, we explored the integrated transcriptional complexity of hippocampi from EPO- and PL-treated mice at a single-nuclei level and, surprisingly, found 20 distinct pyramidal lineages. Using the known markers to classify these neuronal identities, we here provide an unprecedented high-resolution view of the pyramidal repertory, which has clarified many

aspects of neuronal existence. In other words, EPO treatment functioned here as an unexpected tool for discoveries.

First, we deliver an unbiased and comprehensive transcriptional landscape of pyramidal lineages and their pseudo-temporal kinetics during differentiation. Second, we identified previously unspecified cells that seem to be transitional or pre-lineage to the mature neuronal population. Third, we report that the composition is generally maintained between EPO and PL samples, except for a newly formed neuronal cluster enriched dramatically in EPO samples. Fourth, our pairwise comparison of gene expression shows that EPO and PL samples exhibit distinct transcriptomic profiles.

Extending our analysis to reconstruct the GRNs, we observe that the regulons that are critical for long-term memory, combatting depression, improving cognition, and synaptic plasticity are relatively enriched in the newly formed neurons of EPO samples. Of note, newly formed neurons in the hippocampus are the ones that counteract the disorders associated with deteriorated cognition, mood, and memory[74,75,82,83] (also reviewed in ref. [73]). This tallies with our DEG analysis, where the youngest neuronal populations in EPO samples showed the most robust enrichment of genes corresponding to these traits.

Our pseudo-temporal trajectory agrees with the biological route of neuronal differentiation. As anticipated, the newly formed neurons precede the pre-existing ones; unexpectedly, however, multiple newly formed lineages converge into a single mature CA1 neuronal cluster, and we observe significant pseudotime differences upon EPO, within newly formed neuronal lineages. To the best of our knowledge, there has been no prior description of as many lineages of newly formed pyramidal subtypes in the adult hippocampus as we report here[34,84–89]. Deconvolutions of bulk transcriptome from the Hipposeq database[34] argue against them being an artifact of single-nuclei handling or injections (Supplementary Fig. 8a), and the expression of their marker genes also accords with Allen in situ hybridization results[90] (Supplementary Fig. 8b).

Adult neurogenesis in mouse hippocampi is a well-established concept; typically, Tbr1 and Dcx genes are used to identify the newly generated neurons[91–94]. Following these leads, we annotate the lineages with a relatively greater abundance of *Tbr1* and *Dcx* as "*newly formed*" neurons that we presume belong to the early stages of neurogenesis. Besides the expression of *Dcx, Tbr1* and the migrating marker *Reln*, we unravel that these early neuron subtypes could be distinguished based on several other neuronal features. For instance, they express *Sox5*, which is known to keep neurons in an undifferentiated state[40], and vGlut2, supposed to mark the firing ones[43,95,96], or if they are serotonergic, express serotonin receptors at higher level. Based on the bona fide markers[33,34], we could predict if they were positioned in a superficial/deep layer or within the dorsal/ventral axis. Indeed, we do notice that these markers are differentially expressed in one or multiple early lineages, suggesting that EPO influences even formation, migration, and synaptic activity.

Because our study does not present spatial or temporal transcriptomics, we acknowledge that we merely have captured a snapshot of the hippocampus at a fixed stage of development. Henceforth, we do not discard the idea that these neurons are migrating, and their origin remains unclear. Nevertheless, the proposed spatiotemporal path of these neurons is based on the high confidence values gathered from the set of attested studies. Taken together, identifying multiple groups of newly generated neurons and their transcriptional networks might enhance our understanding of the adult neurogenesis processes −partly independent of DNA synthesis− on the molecular and cellular levels. Consistent with earlier studies[17,21,45], we did observe the enhancement of mature CA1 neurons, albeit at a modest level. In contrast, Nf.M.S neurons show a dramatic enrichment of up to 200%. Thus, our analyses independently reproduce the previous findings

whilst providing a trove of new observations in immature neuronal populations.

These dramatic findings naturally raise the question whether the observed formation of new neurons limits the future extent of potential improvement and "hardware upgrade" with long-term rhEPO treatment or whether it would be expected to diminish with age. In fact, rhEPO treatment resulted in both younger and older mice in an increment of mature pyramidal neurons of up to 20%, alleviating concerns of a potential limitation to young age[17,21]. Moreover, temporary decreases in precursor cells upon rhEPO−due to their differentiation to the next stages−were found to have returned to control levels after few weeks[17].

Our GRN analysis signifies the ability of EPO in rewiring the complex regulatory landscape in newly formed neurons, including IEGs and a set of other genes essential in the recovery from neuropsychiatric disorders[67–71,97,98], potentially explaining its beneficial effects in these conditions[76–81]. To generalize these inferences, we ought to have independent validations; nonetheless, this study provides an exciting launchpad for understanding the interplay of EPO with chromatin and transcriptional regulators in healthy and pathological conditions. Intriguingly, various studies have independently reported EPO-mediated induction of IEGs that we here demonstrate to be upregulated in EPO samples from newly formed lineages[99–101]. Resolving this topic further, we show that the induction of IEGs are specific to newly formed lineages (Supplementary Fig. 9), in agreement with the potential of EPO to re-wire cognition-associated transcriptional networks.

Showing that EPO modulates transcriptional activity of neurogenesis and synapse-associated genes, this study establishes profound implications not only for neuronal development but also for disorders characterized by defects in neurodevelopment and synapse organization[102,103]. Besides, the earlier studies have reported a viable causative link between EPO activity and cognitive improvement[14,22–24,77], the connection between EPO and transcriptome changes leading to restructure the synapse was previously overlooked. We uncover how EPO engages in a mutualistic interaction between neuronal differentiation and trans-synaptic activity via re-structuring the GRNs.

With an advanced genetic approach to segregate the newly formed and pre-existing neurons in murine hippocampi for single-cell patch-clamping, we demonstrate remarkable electrophysiological differences between EPO and PL samples for these two neuronal populations. Together, these robust findings should now stimulate a large set of biochemical and genetic manipulation experiments to broaden our appreciation for the constructive force of EPO in reshaping neuronal networks and providing "brain hardware upgrade for escalating performance on demand".

While our survey of neuronal lineages in the hippocampus is the broadest of its kind, it remains limited by the constraints and shortcomings of snRNA-seq. The expression level of any given gene may be underestimated due to dropout effects. In consistence with previous reports on dropout effects of this methodology, our snRNA-seq data did not or hardly detect the expression of EPOR and EPO, respectively[16]. Their expression, however, has been identified employing the more sensitive in situ hybridization[21]. Moreover, RNA expression levels might not be proportional to protein abundance; thus, we ought to corroborate in future studies our observations by quantifying protein expression in situ.

Furthermore, it cannot be entirely excluded that some of the observed difference in a particular set of gene expression is an indirect effect of rhEPO, mediated by the increased number of red blood cells and thus oxygen delivery in rhEPO-treated animals. Even though the direct effect of rhEPO on improved cognition and upregulation of neurotrophic genes is evident from overlaying previous studies with

the present work, additional investigation is warranted to dissect a possible contribution of hematopoiesis to our observed transcriptional changes[14,17,21,101,104,105].

Collectively, we provide a robust and reliable transcriptional landscape of multiple pyramidal lineages ranging from early to late stages of neurogenesis in mouse hippocampus. Initially unintended, EPO treatment served here as a tool for resolving previously unseen pyramidal lineages. We not only disentangle that pre-existing and newly formed neurons co-exist in a distinct transcriptional state but also show that EPO stimulates adult neurodifferentiation and adaptive cellular growth processes that can be harnessed to combat cognitive dysfunction. The comprehensive set of genes whose expression defines the identity of each lineage is invaluable in guiding future studies on neuroplasticity and on the promising role of EPO/EPOR signaling in the treatment of neuropsychiatric disease.

## Methods

All experiments were approved by the local Animal Care and Use Committee (Niedersächsisches Landesamt für Verbraucherschutz und Lebensmittelsicherheit, LAVES) and conducted in accordance with the German Animal Protection Law.

### Experimental model and treatments

C57BL6/N (WT) and NexCreERT2::R26R-tdT male mice received intraperitoneal injections (i.p.) of recombinant human (rh)EPO (5000 IU/kg body weight; NeoRecormon, Roche) or PL (solvent solution, 0.01 ml/g) every other day for 3 consecutive weeks starting on P28. To induce CreERT2 activity in NexCreERT2::R26R-tdT, tamoxifen solution (10 mg/ml) was freshly prepared by dissolving tamoxifen freebase (Sigma) in corn oil (Sigma) at room temperature (RT) for 45 min. Mice received a total of 5 i.p. injections of 100 mg/kg tamoxifen over the course of 3 days starting at P23. At 48 h after the last tamoxifen injection, EPO/PL treatment was initiated at P28.

### Single-nuclei RNA sequencing

On P49, 24 h after the last EPO/PL injection, 23 mice ($N = 11$ EPO, $N = 12$ PL) were sacrificed by cervical dislocation. The brain was immediately removed, without anesthesia, the right hippocampus dissected on an ice-cold plate, quickly immersed in liquid nitrogen, and kept at −80 °C. Two right hippocampi of the same treatment group were collected in one tube for sequencing (one tube in the EPO group with only one right hippocampus). The final analysis was performed on $N = 6$ tubes per group. Single-nucleus suspension was prepared using 10x Genomics Chromium Single Cell 3′ Reagent Kits v3 (10X Genomics, Pleasanton, CA) according to the manufacturer's protocol. Quantity and quality of cDNA were assessed by Agilent 2100 expert High Sensitivity DNA Assay. cDNA samples were sequenced on NovaSeq 6000 S2 flow cell at UCLA Technology Center for Genomics and Bioinformatics.

### Single-nuclei RNA-seq data processing

Sample single-cell fastqs were aligned to the mouse genome (mm10) using 10x Genomics CellRanger count (v6.1.1) to obtain gene/cell count matrices. The respective genome references and gene transfer format (GTF) files were obtained from Ensembl and prepared with CellRanger's mkref function. The alignment was run with standard parameters described in the developer's manual. Afterward, to avoid potential issues with batch effects and in differential gene expression, background RNA was removed using CellBender version 0.2.1[106]. Quality of alignment and data matrices were tested using the downstream processing tools from CellRanger.

Seurat (v4.1.1)[30], implemented in R (v4.1.0)[107], was used for filtering, normalization, and cell-types clustering. The sub-clusters of cell types were annotated based on the known transcriptional markers from the literature survey. Briefly, we performed the following data processing steps: (1) we determined and removed the plausible

doublets using the publically available tool "DoubletFinder"[108]; (2) cells were filtered based on the criteria that individual cells must be expressing at least 500 and not more than 11,200 genes with a count ≥1 (specific maximum value was individually determined for each sample); (3) we normalized and regressed out the impact of counts mapping to mitochondrial genes; (4) data normalization was performed by dividing uniquely mapping read counts (defined by Seurat as unique molecular identified [UMI]) for each gene by the total number of counts in each cell and multiplying by 10,000. These normalized values were then log-transformed. We further normalized the data using ribosomal and cell-cycle genes. Cell types were clustered using the top 2000 variable genes expressed across all samples. Clustering was performed using the "FindClusters" function with essentially default parameters, except the resolution was set to 0.6. The first 30 PCA dimensions were used in constructing the shared-nearest neighbor (SNN) graph and generating 2-dimensional embeddings for data visualization using UMAP. Major cell types were assigned based on the popular markers, and cell subtypes within major cell types were annotated using the sub-cluster markers obtained using default parameters. We then chose the pyramidal lineages to perform the single-nuclei trajectory, pseudotime analysis and cell ordering along an artificial temporal continuum analysis using Monocle2 and Slingshot[50,51]. The top 500 differentially expressed genes were used to distinguish between the sub-clusters of pyramidal populations on pseudotime trajectory. The transcriptome from every single nucleus represents a pseudotime point along an artificial time vector that denotes the progression of mature neurons from the newly formed ones.

To compare the differentially expressed genes between EPO and PL samples, we first transformed the data into a pseudobulk expression matrix by averaging all genes' expression in each cell type. We then performed differentially expressed gene analysis between the two groups of samples using the glmLRT that performs the likelihood ratio method incorporating the uncertainty in the count estimation while calculating the significance of DE detection inbuilt in the edgeR package. For analyzing the GRNs, we used the SCENIC[66] package. This computational strategy uses multiple sub-packages with algorithms required to find out GRNs in every single nucleus. The first step is to find the co-expression networks, based on the genome-wide correlation analysis from the GENIE3 algorithm[66]. It further infers the regulatory networks from expression data using tree-based methods to construct modules. These are the gene sets that are co-expressed with a master regulator, that is inferred using the random forest regression. SCENIC provides several thresholds to build valid modules. We only selected those modules that did not had lncRNAs as the top regulators. We kept significant thresholds of predicted weight for each module regulators (links with weight >0.01). This assisted us to avoid an excess of arbitrary thresholds for detecting regulons containing protein-coding genes as the master regulator. Only the gene sets (co-expression modules) with at least 15 genes were kept for the AUCell scores (with aucMaxRank = 10%). AUC values (quantified by AUCell) are further transformed into a binary activity matrix as suggested by SCENIC. We used either the AUC scores directly for a heatmap, or a binary matrix using a cutoff (determined automatically) of the AUC score.

To calculate the enrichment score of each cell type in EPO and PL samples, we first calculated the fraction of single nuclei in each sample per lineage (Observed) and the proportion of the rest of the single nuclei in each sample per lineage (Expected). The enrichment ratio shown on the plot is the log ratio of Observed and Expected values for each cell type. We further confirmed the enrichment using the linear regression model to decipher the enrichment significance between the analyzed samples. The p value was calculated using a two-sided Fisher exact test followed by Bonferroni correction. Specific codes of data/plots and the way lineages and cell types are classified in our study are available on GitHub link (https://github.com/Manu-1512/Erythropoetin-says-Dracarys).

Finally, we used WebGestalt v. 2019[109] to identify enriched ontology terms using over-representation analysis (ORA). We used ORA to identify enriched terms for three pathway databases (KEGG, Reactome, and Wikipathway), three disease databases (Disgenet, OMIM, and GLAD4U), and human phenotypic database.

## Electrophysiology

Male NexCreERT2:tdTomato mice (7-8 weeks old) were anesthetized with isoflurane and quickly decapitated. The brain was rapidly removed and placed in ice-cold slicing solution consisting of (in mM) 125 NaCl, 2.5 KCl, 26 NaHCO$_3$, 1.25 NaH$_2$PO$_4$, 25 glucose, 4 lactate, 1 MgCl$_2$, 2 CaCl$_2$ and pre-equilibrated with 95% O$_2$/5% CO$_2$. Coronal sections (300 μm thick) were cut by using a vibrating-blade microtome (VT1200s, Leica). During recovery, slices were maintained at near physiological temperature (34-35 °C) in artificial cerebrospinal fluid (aCSF) consisting of (in mM) 125 NaCl, 4 KCl, 26 NaHCO$_3$, 10 glucose, 1.3 MgCl$_2$, 2 CaCl$_2$, which was continuously bubbled with 95% O$_2$/5% CO$_2$.

After a recovery period of ≥45 min, slices were transferred to a recording chamber and continuously superfused with aCSF at a rate of 1–2 ml/min. CA1 pyramidal neurons were visually identified using a Zeiss Axio Examiner D1 microscope equipped with Dodt Gradient Contrast Optics (Zeiss, Dodt for condenser NA 0.9) and a 40x water-immersion objective (Zeiss, W Plan-Apochromat 40×/1.0 DIC VIS-IR). Pre-existing and newly formed neurons were distinguished by the presence or absence of tdTomato fluorescence, respectively, which was examined by using a fluorescent light source (Zeiss, HXP 120 v) and an appropriate filter set (ex 560/40, FT 585, em 630/75). Live images were captured via a microscope-mounted camera (Zeiss Axiocam 503 mono) coupled to a PC running Zen 2.3 imaging software (Zeiss).

Whole-cell patch-clamp recordings were obtained at room temperature (21–22 °C) with an EPC10 amplifier (HEKA Electronic, Germany), controlled by Patchmaster software (HEKA Electronic, Germany). Patch electrodes (2.5–5 MΩ open tip resistance when filled with intracellular solution) were pulled from borosilicate glass capillaries. To estimate the cell capacitance, hyperpolarizing voltage steps of 10 ms duration (from V$_h$ = −70 to −80 mV) were delivered immediately after formation of the whole-cell configuration. Capacitance values reported represent total cell capacitance calculated according to a two-compartment equivalent circuit model[110]. Input resistance was estimated from the slope conductance of I-V plots which were constructed from membrane potential changes measured in response to small current injections under current-clamp conditions. The identity of visually identified CA1 pyramidal cells was confirmed based on their passive membrane properties and their discharge behavior in response to depolarizing current steps.

To record miniature excitatory postsynaptic currents (mEPSC) or miniature inhibitory postsynaptic currents (mIPSC), action potential firing was suppressed by adding 1 μM tetrodotoxin (TTX, Tocris) to the bath solution. mEPSC were pharmacologically isolated by blocking GABAergic IPSCs (50 μM picrotoxin, Tocris) and were recorded using an internal solution consisting of (in mM) 126 K-gluconate, 4 KCl, 10 HEPES, 0.1 EGTA, 4 Mg-ATP, 10 Phosphocreatine, 0.3 Na-GTP. mIPSC were pharmacologically isolated by blocking glutamatergic EPSCs (2 μM NBQX and 2 μM CPP, Tocris) and were recorded using an internal solution consisting of (in mM) 4 K-gluconate, 130 KCl, 10 HEPES, 0.1 EGTA, 4 Mg-ATP, 10 Phosphocreatine, 0.3 Na-GTP. Both mEPSC and mIPSC were recorded under whole-cell voltage-clamp using a holding potential of −70 mV. Discharge properties of pyramidal neurons were characterized under current-clamp conditions. Action potentials (AP) were evoked with depolarizing current injections in 50 pA amplitude increments. The AP firing threshold of neurons was estimated from AP phase-plane plots[111]. Recordings with access resistance >20 MΩ or leak current >200 pA were rejected from the analysis. The data were sampled at 20 kHz and low-pass filtered at cutoff frequency of 5 kHz. Off-line data analysis was performed in IgorPro (Wavemetrics, USA). Data are presented as mean ± SEM and statistical significance of mean differences was determined by two-way ANOVA with Tukey's method for multiple comparison correction.

## Reporting summary

Further information on research design is available in the Nature Portfolio Reporting Summary linked to this article.

## Data availability

Raw and processed snRNA-seq data are publicly available on GEO via accession code GSE220522. Source data are provided with this paper.

## Code availability

Computer code is available from GitHub under https://github.com/Manu-1512/Erythropoetin-says-Dracarys, also provided in the Source Data file and linked to Zenodo repository[112] cited as https://doi.org/10.5281/zenodo.8071471.

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

## Acknowledgements

This work has been supported by the European Research Council (ERC) Advanced Grant to HE under the European Union's Horizon Europe research and innovation program (acronym *BREPOCI;* grant agreement No 101054369), as well as by the Max Planck Society, the Max Planck Förderstiftung, the Deutsche Forschungsgemeinschaft (DFG, German Research Foundation), via DFG-Center for Nanoscale Microscopy & Molecular Physiology of the Brain (CNMPB) and DFG-TRR 274/1 2020—408885537. V.D.G. received support from the IMPRS-Genome Science PhD program. Y.C. is a recipient of a grant from the Peter and Traudl Engelhorn Foundation. The authors are grateful to Qing Wang, UCLA, who performed the sample preparation necessary for the sequencing in our study.

## Author contributions

Concept, design, and supervision: M.S., H.E., K.A.N., R.K., N.B., and D.G. Drafting the manuscript and display items: M.S., V.D.G., S.W., Y.Z., and Y.C., together with H.E. Data acquisition/generation: R.K., D.G., Y.Z., and L.F.G. Data analyses and interpretation: M.S., Y.Z., V.D.G., S.W., Y.C., R.M.M., H.T., D.G., K.A.N., and H.E. All authors read and approved the final version of the manuscript.

## Funding

## Competing interests

The authors declare no competing interests.
