## [Peer Review File · Nature Communications]

Erythropoietin re-wires cognition-associated transcriptional networksREVIEWER COMMENTS

Reviewer #1 (Remarks to the Author):

Singh et al. address the precognitive effects associated with recombinant human erythropoietin. The authors previously reported that EPO treatment increased maturation of non-proliferating precursors, enhanced dendritic spine density, and increased CA1 pyramidal neurons of up to 20% without proliferation accompanied by improved motor-cognitive performance. In this report, mice treated with EPO or control are used for transcriptional hippocampal profiling of 108,000 single nuclei to identify changes in cell populations. Transcriptome analysis provided a broader landscape of mouse hippocampus at the single nuclei resolution than appreciated previously. The subset of ~36,000 pyramidal nuclei, defined 20 distinct cellular clusters by a unique set of marker genes. EPO treatment upregulated genes crucial for neurodifferentiation, dendrite growth, synaptogenesis, memory formation, and cognition. A cluster identified here as 'Newly formed-migrating-superficial' (Nf.M.S) pyramidal neurons showed a dramatic enrichment of up to 200% with EPO treatment and are derived from multiple predecessor lineages. A genetic approach segregated newly formed and pre-existing neurons for single-cell patch-clamping which demonstrated electrophysiological differences between EPO and control samples for these two neuronal populations. These data provide mechanistic insight into how EPO modulates neuronal function. Limitations of this study are that the analysis captures hippocampal nuclei at a single stage, does not provide spatial or temporal resolution, or address whether EPO promotes cell migration. Transcriptional differences among the various cell lineages identified here are not verified by corresponding changes in protein expression and are not shown to relate to changes in brain resulting from hypoxia induction of endogenous EPO.

Concerns:

No evidence is provided to indicate that the changes observed here after EPO treatment for 3 weeks are hematopoiesis-independent. Authors should discuss how improved oxygen delivery to brain resulting from increase in red blood cell production could contribute to the changes in transcriptome analysis observed here.

Is the single nuclei transcriptional profiling able to provide information about EPO receptor expression?

Is EPO receptor preferentially expressed in select lineages or cell subsets? Do differences in EPO receptor expression relate to differential excitatory response to EPO treatment?

The authors propose that with EPO treatment, new neurons are formed and pre-mature neurons are diminished. The authors should discuss whether this limits the extent of potential improvement and “hardware upgrade” with long term EPO treatment or whether the potential increase in new neuron formation with EPO treatment would be expected to diminish with age.

Reviewer #2 (Remarks to the Author):

The pleiotropic activities of erythropoietin (EPO) outside of hematopoiesis have been of widespread research interest for several decades. Strong effects of EPO on cognitive function was observed early on

and has been a prominent focus in the research community, to which the senior author's group has contributed significantly. In spite of extensive investigation, the details of this biology have been relatively unknown. The current contribution by Singh and co-investigators represents an important step forward in the field. A detailed methodology is presented which is state of the art and sound. The results based on gene expression analyses show that there is a surprisingly diverse hippocampal pyramidal cell population, a subpopulation of which on exposure to EPO undergo differentiation. There is an upregulation of genes associated with dendrite formation and synaptogenesis, processes of crucial relevance for cognition. Importantly, the observed gene expression differences are followed up by direct electrophysiologic evaluation of new versus old neurons which documents differences which support the conclusions. The very extensive data set will be of significant utility for the further study of hippocampal physiology relevant for development, learning and memory.

There are some specific issues which would benefit from clarification and/or comment.

1) High dose human EPO was administered on alternate days for 3 weeks. This is sufficient time to significantly increase red cell number, potentially providing metabolic effects, e.g., increased oxygen delivery to regions of the brain under "functional (work- or development-related) hypoxia". Functional hypoxia is a crucial process identified previously by this group which generates local production of EPO with subsequent effects on brain cells, as they discuss in the beginning of the Introduction. Additionally, exogenous EPO as administered will itself potentially suppress local EPO production. Do they have information about the hematocrit and can comments be made concerning potential effects of silencing local, physiologic EPO expression? (This impacts potentially with comment 2 below.) An additional possible area of concern in the experimental design is the potential development of anti-human EPO antibodies. These could provoke an immune response within the time frame of the experiments, and if neutralizing, could result in anemia-related hypoxic stress.

2) The period of EPO administration was between 4-7 weeks of age during ongoing maturation of the nervous system. Although 8 weeks is widely considered to be adult, recent data shown developmental changes occur as late as 6 months of age. The relevance of EPO effects during development may be different from those occurring in a fully mature nervous system (for which the cognitive effects of EPO have been observed to occur). As the authors acknowledge, the data obtained in this investigation correspond to a "snap shot" in time. It would be interesting to determine whether key findings also apply to older mice. Authors may wish to comment on this, particularly concerning effects of EPO on developing brain.

3) The observed neuronal changes appear numerically to be surprisingly small considering the effects previously reported in the literature. The authors may wish to comment on this from the perspective of prior work.

4) There is no consideration of presence/changes of a brain receptor for EPO. Presumably these data exist in the data set, and if so, would deserve comment.

5) There are a multitude of abbreviations. A supplemental table listing all of these would be very helpful.

There are some issues with the figures:

Fig. 1 is helpful, but would benefit from a short legend which recapitulates what is portrayed, especially the electrophysiology.

Fig 2: classification was carried out by pooling both the control and treated animals. The analysis was constructed by integrating separate samples (Suppl Fig 1) but were the cellular compositions between the two groups compared? Although this work focuses on pyramidal neurons, it would be helpful to know whether there are differences in other cellular populations (e.g., neuroimmune cells, glia, etc) which may be affected to EPO exposure, as would be predicted by prior work.

Legend of Fig 4 refers to “differentially expressed genes between pyramidal lineages shown in Fig 4b.” Should this not be Fig 3b?

Legend Fig 5 c refers to Fig. 4-5a. This is unclear to me. Is this referring to Fig 5b?

Fig 7b: is the difference between old/new EPO neurons significant?

Suppl Fig 3: is there a key for the group numbers shown in the EPO+placebo group? Suppl Fig 7 has annotations relevant for Suppl Fig 3, but these are color coded (the colors differ between the figures) and are not enumerated. Please harmonize the figures.

Suppl Fig 4: individual data points are not visualized. Do any of these regions differ significantly?

Suppl Fig 5: One panel is unlabeled. In panel a, what are the units of the y axis?

Suppl Fig 7b: The number of DEGs are shown on the side, not the bottom of the heatmap as stated in the legend.

Reviewer #3 (Remarks to the Author):

The authors present an interesting study exploring the transcriptional landscape of hippocampi of mice treated with recombinant EPO. They classify the major neuronal and non-neuronal cell types from the murine hippocampus using snRNA-seq datasets from EPO treated versus untreated mice and found several distinct pyramidal lineages. They also show that recombinant PO modulates transcriptional activity of neurogenesis and synapse-associated genes. Furthermore, they use a single-cell electrophysiology approach to demonstrate that recombinant EPO affects excitatory and inhibitory input to both newly formed and pre-existing pyramidal neurons in mouse hippocampi, with newly formed neurons receiving more excitation and less inhibition under EPO treatment than pre-existing neurons. They suggest the potential clinical benefit of EPO treatment. Overall the EPO-dependency of these effects represents a substantial step forward.

This is an original and well-written study. The methods are sound and the experimental work supports the conclusions and claims, however this reviewer would like to see systematic independent validations of selected subsets of differentially regulated genes (e.g. quantitative PCR-based).

POINT-BY-POINT RESPONSE to the REVIEWERS' COMMENTS

(Reviewers' comments in Calibri 11; **response in ARIAL 12 – bold, blue; quoting the text from the manuscript – italic, regular blue; reference to comment-relevant passages and changes in the manuscript labelled in yellow**)

Reviewer #1:

Singh et al. address the precognitive effects associated with recombinant human erythropoietin. The authors previously reported that EPO treatment increased maturation of non-proliferating precursors, enhanced dendritic spine density, and increased CA1 pyramidal neurons of up to 20% without proliferation accompanied by improved motor-cognitive performance. In this report, mice treated with EPO or control are used for transcriptional hippocampal profiling of 108,000 single nuclei to identify changes in cell populations. Transcriptome analysis provided a broader landscape of mouse hippocampus at the single nuclei resolution than appreciated previously. The subset of ~36,000 pyramidal nuclei, defined 20 distinct cellular clusters by a unique set of marker genes. EPO treatment upregulated genes crucial for neurodifferentiation, dendrite growth, synaptogenesis, memory formation, and cognition. A cluster identified here as 'Newly formed-migrating-superficial' (Nf.M.S) pyramidal neurons showed a dramatic enrichment of up to 200% with EPO treatment and are derived from multiple predecessor lineages. A genetic approach segregated newly formed and pre-existing neurons for single-cell patch-clamping which demonstrated electrophysiological differences between EPO and control samples for these two neuronal populations. These data provide mechanistic insight into how EPO modulates neuronal function. Limitations of this study are that the analysis captures hippocampal nuclei at a single stage, does not provide spatial or temporal resolution, or address whether EPO promotes cell migration. Transcriptional differences among the various cell lineages identified here are not verified by corresponding changes in protein expression and are not shown to relate to changes in brain resulting from hypoxia induction of endogenous EPO.

We thank the Reviewer for elegantly summarising and explicitly pointing out the strength and limitations that we stipulated in our manuscript and for the insightful and constructive feedback. We are glad that our messages went through accurately.

Concerns:

No evidence is provided to indicate that the changes observed here after EPO treatment for 3 weeks are hematopoiesis-independent. Authors should discuss how improved oxygen delivery to brain resulting from increase in red blood cell production could contribute to the changes in transcriptome analysis observed here.

We are glad that the Reviewer has pointed it out, as we should have discussed it in the original version of the manuscript. This concern is partly based on the assumption that the shown transcriptome-wide differences might associate with high hematocrit levels in rhEPO animals. Because we know this could be a factor, we did not claim in the main body of the manuscript that it is hematocrit-independent. The Reviewer points out the first sentence of our abstract. We wrote this to set the stage for the work presented here. This is the only instance where we mention “*hematopoiesis-independent*”. Since it was a section of the abstract, the relevant research was not cited. We have revised the corresponding sentence to add extra cautionary language to that effect (now the sentence reads as “*Recombinant human erythropoietin (rhEPO) has potent procognitive effects, likely hematopoiesis-independent, but underlying mechanisms and physiological role of brain-expressed EPO remained obscure*”).

We, however, have previously shown that the knock-out of Epor, specifically in pyramidal lineages, reversed the changes in the hippocampus and overall behaviour that was attained by rhEPO treatment (*Wakhloo et al 2020*). Moreover, the transgenic overexpression of Epor in hippocampus pyramidal neurons also had enhanced memory and cognition (*Sargin et al 2011*). While we agree with the Reviewer that high numbers of red blood cells and oxygen delivery in the brain might alter the transcriptome, the above results show that it is not indispensable for the phenotypes we observed. Importantly, first effects on the transcriptome of pyramidal neurons are already observed after 6h of a single intraperitoneal injection of EPO, at a time when no changes in hematocrit are seen (*Wakhloo et al 2020*).

References:

Wakhloo, D. et al. Functional hypoxia drives neuroplasticity and neurogenesis via brain erythropoietin. *Nat Commun* 11, 1313 (2020).

Sargin, D. et al. Expression of constitutively active erythropoietin receptor in pyramidal neurons of cortex and hippocampus boosts higher cognitive functions in mice. *BMC Biol* 9, 27 (2011).

Pushing this frontier further, a wealth of data advocates the hematocrit-independent functions of EPO in the brain. See, for instance:

Schuler, B., Vogel, J., Grenacher, B., Jacobs, R. A., Arras, M. & Gassmann, M. Acute and chronic elevation of erythropoietin in the brain improves exercise performance in mice without inducing erythropoiesis. *The FASEB Journal* 26, 3884-3890 (2012).

We demonstrate here that the significant transcriptome changes in newly formed lineages in EPO subjects revolve around the regulatory networks of memory, mood and cognition-associated genes such as *Bdnf*, *Arc*, *Nr4a1*, *Egr1,2,3* etc. Interestingly, it was shown that the expression of these genes was induced upon carbamoylated erythropoietin (CEPO), a modified version of EPO which does not induce erythropoiesis (*Tiwari et al 2019; Leist et al 2004*), confirming the hematopoiesis-independent mode of its action. These observations convincingly suggest that as far as neurotrophic genes are concerned, the observed transcriptional changes might be direct effects of rhEPO.

We agree with the Reviewer that we must discuss the impact of hematocrit and oxygen delivery to better clarify our findings. To clarify and follow the Reviewer's suggestion, we now address this issue in the discussion section **page 15**, which reads as follows:

'Furthermore, it cannot be entirely excluded that some of the observed difference in a particular set of gene expression is an indirect effect of rhEPO, mediated by the increased number of red blood cells and thus oxygen delivery in rhEPO treated animals. Even though the direct effect of rhEPO on improved cognition and upregulation of neurotrophic genes is evident from overlaying previous studies with the present work (references below), additional investigation is warranted to dissect a possible contribution of hematopoiesis to observed transcriptional changes.'

References:

Wakhloo, D. et al. Functional hypoxia drives neuroplasticity and neurogenesis via brain erythropoietin. *Nat Commun* 11, 1313 (2020).

Sargin, D. et al. Expression of constitutively active erythropoietin receptor in pyramidal neurons of cortex and hippocampus boosts higher cognitive functions in mice. *BMC Biol* 9, 27 (2011).

Tiwari, N. K., Sathyanesan, M., Schweinle, W. & Newton, S. S. Carbamoylated erythropoietin induces a neurotrophic gene profile in neuronal cells. *Prog Neuropsychopharmacol Biol Psychiatry* 88, 132-141 (2019).

Leist, M. et al. Derivatives of Erythropoietin That Are Tissue Protective But Not Erythropoietic. *Science* 305, 239-242 (2004).

Adamcio, B. et al. Erythropoietin enhances hippocampal long-term potentiation and memory. *BMC Biol* 6, 37 (2008).

Hassouna, I. et al. Revisiting adult neurogenesis and the role of erythropoietin for neuronal and oligodendroglial differentiation in the hippocampus. *Mol Psychiatry* 21, 1752-1767 (2016).

Is the single nuclei transcriptional profiling able to provide information about EPO receptor expression? Is EPO receptor preferentially expressed in select lineages or cell subsets? Do differences in EPO receptor expression relate to differential excitatory response to EPO treatment?

The Reviewer raises a crucial point. Dropout effects are a well-known, general problem in sc/snRNA-seq. As reported, one of the shortcomings of our hippocampal sc/snRNA-seq data are dropout effects on the expression of EPO and EPOR – see present paper and explicit mentioning in:

Ehrenreich et al Introducing the brain erythropoietin circle to explain adaptive brain hardware upgrade and improved performance. *Mol Psychiatry*. 2022 May;27(5):2372-2379. doi: 10.1038/s41380-022-01551-5. Epub 2022 Apr 12. PMID: 35414656; PMCID: PMC9004453.

In addition, this important point has been addressed via in situ hybridization experiments in our previously published work (*Wakhloo et al 2020*).

EPO and EPOR are known to be lowly expressed in general, and in brain in particular, and/or may exhibit a non-nuclear localization. We have stated this dropout issue in our discussion section, which we have now extended, page 15:

'While our survey of neuronal lineages in the hippocampus is the broadest of its kind, it remains limited by the constraints and shortcomings of snRNA-seq. The expression level of any given gene may be underestimated due to dropout effects. In consistence with previous reports on dropout effects of this methodology, our snRNA-seq data did not or hardly detect the expression of EPOR and EPO, respectively (*Ehrenreich et al 2022*). Their expression, however, has been identified employing the more sensitive in situ hybridization (*Wakhloo et al 2020*). Moreover, RNA expression levels might not be proportional to protein abundance; thus, we ought to corroborate in future studies our observations by quantifying protein expression in situ.'

The expression dynamics of EPOR in the murine hippocampus have been shown in several studies from independent laboratories, using in situ hybridization and immunohistochemistry, e.g. references below. EPOR is widely expressed in the hippocampus, mainly by oligodendrocytes and pyramidal neurons. Among other cell types, interneurons, astrocytes, microglia, and endothelial cells also express EPOR, albeit lower than pyramidal neurons and oligodendrocytes. Although our snRNA-seq data does not resolve the expression of EPOR completely and precisely in cell clusters due to the dropout effects, we still can see its (low) expression in these cell types. Of note, the endogenous EPOR expression in interneurons has not been shown before and our follow-up study (soon to be submitted) will report its expression in different interneuronal subpopulations using different methodological approaches.

Indeed, EPOR expression directly associates with the excitability of pyramidal neurons that are affected by EPO treatment. We had shown that the pyramidal dendritic spine density that was enhanced due to EPO treatment, was abolished upon depleting EPOR (*Wakhloo et al 2020*). These genetic perturbation experiments are sufficient to suggest a direct involvement of EPOR in the EPO-mediated excitability of pyramidal neurons.

References:

Wakhloo, D. et al. Functional hypoxia drives neuroplasticity and neurogenesis via brain erythropoietin. *Nat Commun* 11, 1313 (2020).

Sargin, D. et al. Expression of constitutively active erythropoietin receptor in pyramidal neurons of cortex and hippocampus boosts higher cognitive functions in mice. *BMC Biol* 9, 27 (2011).

Adamcio, B. et al. Erythropoietin enhances hippocampal long-term potentiation and memory. *BMC Biol* 6, 37 (2008).

Fernandez Garcia-Agudo, L. et al. Brain erythropoietin fine-tunes a counterbalance between neurodifferentiation and microglia in the adult hippocampus. *Cell Rep* 36, (2021).

Liu, C., Shen, K., Liu, Z. & Noguchi, C. T. Regulated Human Erythropoietin Receptor Expression in Mouse Brain. *Journal of Biological Chemistry* 272, 32395–32400 (1997).

Ott, C. et al. Widespread Expression of Erythropoietin Receptor in Brain and Its Induction by Injury. *Mol Med* 21, 803–815 (2015).

Eid, T. et al. Increased Expression of Erythropoietin Receptor on Blood Vessels in the Human Epileptogenic Hippocampus with Sclerosis. *J Neuropathol Exp Neurol* 63, 73–83 (2004).

Khalid, K. et al. Erythropoietin Stimulates GABAergic Maturation in the Mouse Hippocampus. *eNeuro* 8, ENEURO.0006-21.2021 (2021).

The authors propose that with EPO treatment, new neurons are formed and pre-mature neurons are diminished. The authors should discuss whether this limits the extent of potential improvement and “hardware upgrade” with long term EPO treatment or whether the potential increase in new neuron formation with EPO treatment would be expected to diminish with age.

We agree that this is a highly interesting aspect. Although it is not mentioned in the present paper, we have not observed any nominal phenotypic differences while dealing with older or younger mice. We have convincingly shown this in a previous paper (*Wakhloo et al 2020*). Our results demonstrated there that the increment in pyramidal neurons and dendritic spine density was consistent and comparable in both younger and older mice.

Following the Reviewer’s comment, we now added respective sentences to the Discussion to clarify and summarise our inferences as to the age and long term effects of EPO treatment. The newly added sentences, page 14 read as follows:

‘These dramatic findings naturally raise the question whether the observed formation of new neurons limits the future extent of potential improvement and ‘hardware upgrade’ with long-term rhEPO treatment or whether it would be expected to diminish with age. In fact, rhEPO treatment resulted in both younger and older mice in an increment of mature pyramidal neurons of up to 20%, alleviating concerns of a potential limitation to young age (*Wakhloo et al, 2020; Hassouna et al 2016*). Moreover, temporary decreases in precursor cells upon rhEPO – due to their differentiation to the next stages – were found to have returned to control levels after few weeks (*Hassouna et al 2016*).’

References:

Wakhloo, D. et al. Functional hypoxia drives neuroplasticity and neurogenesis via brain erythropoietin. *Nat Commun* 11, 1313 (2020).

Hassouna, I. et al. Revisiting adult neurogenesis and the role of erythropoietin for neuronal and oligodendroglial differentiation in the hippocampus. *Mol Psychiatry* 21, 1752–1767 (2016).

Reviewer #2:

The pleiotropic activities of erythropoietin (EPO) outside of hematopoiesis have been of widespread research interest for several decades. Strong effects of EPO on cognitive function was observed early on and has been a prominent focus in the research community, to which the senior author's group has contributed significantly. In spite of extensive investigation, the details of this biology have been relatively unknown. The current contribution by Singh and co-investigators represents an important step forward in the field. A detailed methodology is presented which is state of the art and sound. The results based on gene expression analyses show that there is a surprisingly diverse hippocampal pyramidal cell population, a subpopulation of which on exposure to EPO undergo differentiation. There is an upregulation of genes associated with dendrite formation and synaptogenesis, processes of crucial relevance for cognition. Importantly, the observed gene expression differences are followed up by direct electrophysiologic evaluation of new versus old neurons which documents differences which support the conclusions. The very extensive data set will be of significant utility for the further study of hippocampal physiology relevant for development, learning and memory.

We thank the Reviewer for the time invested in carefully reviewing our manuscript and the highly positive feedback. We appreciate your support and constructive assessment.

There are some specific issues which would benefit from clarification and/or comment.

1) High dose human EPO was administered on alternate days for 3 weeks. This is sufficient time to significantly increase red cell number, potentially providing metabolic effects, e.g., increased oxygen delivery to regions of the brain under "functional (work- or development-related) hypoxia". Functional hypoxia is a crucial process identified previously by this group which generates local production of EPO with subsequent effects on brain cells, as they discuss in the beginning of the Introduction. Additionally, exogenous EPO as administered will itself potentially suppress local EPO production. Do they have information about the hematocrit and can comments be made concerning potential effects of silencing local, physiologic EPO expression? (This impacts potentially with comment 2 below.)

Allow us to consider both concerns in a turn.

a) Do they have information about the hematocrit ?

We have extensive information accumulated over the years: One example is Adamcio et al 2008: 'Whereas at 1 week after termination of the 3-week treatment, hematocrit was still increased in EPO-treated mice (control mice: $36.5 \pm 0.84\%$, $N = 8$; EPO mice: $53.3 \pm 1.34\%$, $N = 10$; $P < 0.0001$)...'

Adamcio B, Sargin D, Stradomska A, Medrihan L, Gertler C, Theis F, Zhang M, Müller M, Hassouna I, Hanneke K, Sperling S, Radyushkin K, El-Kordi A, Schulze L, Ronnenberg A, Wolf F, Brose N, Rhee JS, Zhang W, Ehrenreich H. Erythropoietin enhances hippocampal long-term potentiation and memory. BMC Biol. 2008 Sep 9;6:37. doi: 10.1186/1741-7007-6-37. PMID: 18782446; PMCID: PMC2562991.

We also have a more recent piece of data about the hematocrit in rhEPO-treated DTA and control mice (still unpublished). As expected, 3-week rhEPO treatment did cause an increase in hematocrit right after treatment cessation. We have now prepared for the reviewer's interest the new figure on the right.

Tam = 2x Tamoxifen (induces DTA expression and thereby gray matter inflammation in our transgenic mice); Oil = Corn oil control (healthy mice).

In particular, we are aware that high blood cells in the brain might alter the transcriptome, and this exact concern was raised by Reviewer 1 (see response above). We have now addressed this point in our Discussion section page 15:

'Furthermore, it cannot be entirely excluded that some of the observed difference in a particular set of gene expression is an indirect effect of rhEPO, mediated by the increased number of red blood cells and thus oxygen delivery in rhEPO treated animals. Even though the direct effect of rhEPO on improved cognition and upregulation of neurotrophic genes is evident from overlaying previous studies with the present work, additional investigation is warranted to dissect a possible contribution of hematopoiesis to our observed transcriptional changes.'

References:

Wakhloo, D. et al. Functional hypoxia drives neuroplasticity and neurogenesis via brain erythropoietin. *Nat Commun* 11, 1313 (2020).

Sargin, D. et al. Expression of constitutively active erythropoietin receptor in pyramidal neurons of cortex and hippocampus boosts higher cognitive functions in mice. *BMC Biol* 9, 27 (2011).

Tiwari, N. K., Sathyanesan, M., Schweinle, W. & Newton, S. S. Carbamoylated erythropoietin induces a neurotrophic gene profile in neuronal cells. *Prog Neuropsychopharmacol Biol Psychiatry* 88, 132–141 (2019).

Leist, M. et al. Derivatives of Erythropoietin That Are Tissue Protective But Not Erythropoietic. *Science* 305, 239–242 (2004).

Adamcio, B. et al. Erythropoietin enhances hippocampal long-term potentiation and memory. *BMC Biol* 6, 37 (2008).

Hassouna, I. et al. Revisiting adult neurogenesis and the role of erythropoietin for neuronal and oligodendroglial differentiation in the hippocampus. *Mol Psychiatry* 21, 1752–1767 (2016).

- b) can comments be made concerning potential effects of silencing local, physiologic EPO expression? (This impacts potentially with comment 2 below.)

This is certainly a valid and potentially important point which should be specifically studied in the future. Admittedly, we are unaware of any work clearly showing an influence of exogenously administered rhEPO on endogenous EPO expression.

But here is - for the Reviewer's interest - some evidence that points against such an influence: In our clinical study in schizophrenia, we measured serum EPO levels over 12 weeks of weekly treatment with high-dose rhEPO (blood taken always immediately before the next infusion, i.e. one week after the last rhEPO infusion) and found that levels did not differ among groups (EPO versus placebo) upon inclusion and remained in the same range within both groups over the whole study period, independent of treatment. In other words, these data did not provide any evidence of suppression of endogenous EPO levels by high-dose rhEPO treatment.

Ehrenreich H, Hinze-Selch D, Stawicki S, Aust C, Knolle-Veentjer S, Wilms S, Heinz G, Erdag S, Jahn H, Degner D, Ritzen M, Mohr A, Wagner M, Schneider U, Bohn M, Huber M, Czernik A, Pollmächer T, Maier W, Sirén AL, Klosterkötter J, Falkai P, Rütther E, Aldenhoff JB, Krampe H. Improvement of cognitive functions in chronic schizophrenic patients by recombinant human erythropoietin. *Mol Psychiatry*. 2007 Feb;12(2):206-20. doi: 10.1038/sj.mp.4001907. Epub 2006 Oct 10. PMID: 17033631.

An additional possible area of concern in the experimental design is the potential development of anti-human EPO antibodies. These could provoke an immune response within the time frame of the experiments, and if neutralizing, could result in anemia-related hypoxic stress.

We assume that the Reviewer refers to the anti-EPO AB formation found in the early 2000s in humans upon rubber stopper in syringes, acting as Freund's adjuvant. This made people strongly aware of this (rare) possibility and AB formation has also been observed in single animals upon treatment with rhEPO.

However, the lead symptom caused by neutralizing antibodies would be anemia, as the Reviewer points out and none of our mice were anemic.

Nevertheless, we asked if immune-related or stress-responsive genes are activated in the EPO samples. We also catalogued the list of expressed and differentially expressed genes (DEGs) in each lineage, shown in Figure 1 of our manuscript. We did not find any enrichment of immune or stress response genes in any of the analysed lineages. To further address this issue, we tested the expression of key marker genes for immune cell activation between EPO and placebo samples. We found that none of these genes showed a detectable expression in any of the samples except for *Nfkb1*, which, showed a similar pattern of expression in EPO and placebo samples. **Figures below for the Reviewer's interest:**

2) The period of EPO administration was between 4-7 weeks of age during ongoing maturation of the nervous system. Although 8 weeks is widely considered to be adult, recent data shown developmental changes occur as late as 6 months of age. The relevance of EPO effects during development may be different from those occurring in a fully mature nervous system (for which the cognitive effects of EPO have been observed to occur). As the authors acknowledge, the data obtained in this investigation correspond to a “snap shot” in time. It would be interesting to determine whether key finding also apply to older mice. Authors may wish to comment on this, particularly concerning effects of EPO on developing brain.

This Reviewer’s suggestion is to comment on the impact of EPO on the developing brain and older mice. Reviewer 1 also suggested discussing the age-related effects of EPO. We posit that our response to the question asked by Reviewer 1 will also suffice this Reviewer’s concern:

We agree that this is a highly interesting aspect. Although it is not mentioned in the present paper, we have not observed any nominal phenotypic differences while dealing with older or younger mice. We have convincingly shown this in a previous paper (*Wakhloo et al 2020*). Our results demonstrated there that the increment in pyramidal neurons and dendritic spine density was consistent and comparable in both younger and older mice.

Following the Reviewer’s comment, we now added respective sentences to the Discussion to clarify and summarise our inferences as to the age effects of EPO treatment. The newly added sentences, **page 14 read now:**

‘These dramatic findings naturally raise the question whether the observed formation of new neurons limits the future extent of potential improvement and ‘hardware upgrade’ with long-term rhEPO treatment or whether it would be expected to diminish with age. In fact, rhEPO treatment resulted in both younger and older mice in an increment of mature pyramidal neurons of up to 20%, alleviating concerns of a potential limitation to young age (*Wakhloo et al, 2020; Hassouna et al 2016*). Moreover, temporary decreases in precursor cells upon rhEPO – due to their differentiation to the next stages – were found to have returned to control levels after few weeks (*Hassouna et al 2016*).

References:

Wakhloo, D. et al. Functional hypoxia drives neuroplasticity and neurogenesis via brain erythropoietin. *Nat Commun* 11, 1313 (2020).

Hassouna, I. et al. Revisiting adult neurogenesis and the role of erythropoietin for neuronal and oligodendroglial differentiation in the hippocampus. *Mol Psychiatry* 21, 1752–1767 (2016).

3) The observed neuronal changes appear numerically to be surprisingly small considering the effects previously reported in the literature. The authors may wish to comment on this from the perspective of prior work.

This is perhaps a misunderstanding; we should have used more transparent language to convey the drawn inferences. In fact, this study reinforces the previous observations to a greater extent. The numerical values in Figure 3E appear smaller because they are illustrated as fold changes for each lineage, i.e., scaled values compared to all the lineages pooled together. To clarify this paradox, we also show the numerical difference in the per cent of these lineage compositions, which is more significant than previously appreciated (Supplementary Figure5). For instance, N.F.M.S neuronal lineage is around 4-5% in placebo samples, whereas EPO samples harbour it up to 10-12%. The numerical calculations of scaled fold change are around +2 in EPO and -2 in

placebo, which obviously appears numerically smaller. Still, it reflects the increment up to 200% of N.F.M.S. in EPO-treated animals. Considering the prior work, for the mature CA1 neurons that were previously shown to increase by 20% (Wakhloo et al 2020), the data presented in the current study, too, find its 15-20 % increment in EPO samples using the same mode of calculations.

Wakhloo, D. et al. Functional hypoxia drives neuroplasticity and neurogenesis via brain erythropoietin. Nat Commun 11, 1313 (2020).

We hope, the Reviewer is not pointing towards differentially expressed genes in each lineage. These numbers are also not small. N.F.M.S showed around 1000 DEGs upon comparing 5000-8000 total genes. These gene lists are obtained using the most robust criteria to calculate differentially expressed genes.

To further accentuate, we edited both figure legends. In Figure 3E, we add: “Heatmap illustrating the relative abundance of each pyramidal cell type shown in Fig. 3b in EPO and PL samples. The relative abundance was calculated by determining the observed fraction of each cell-type compared to the expected fraction and using a two-sided Fisher exact test to identify cell types that were significantly enriched in EPO samples. Stars denote the p-values that have been adjusted using the Bonferroni correction ($p < 0.05 = *$, $p < 0.01 = **$, $p < 2.2e-16 = ***$). Of note, the changes in neuronal composition are scaled values compared to all the lineages pooled together.”

Notably, in Figure 6A, we replaced the test name to be more explicit. “Volcano plot showing genes that are differentially expressed between EPO and PL samples in newly formed and mature CA1 neurons. The horizontal dashed line indicates $-\text{Log}_{10}P = 2$ (p-value, adjusted using the glmLRTTest). Boxed text beside the volcano plot corresponds to gene ontologies in which genes that are differentially expressed between EPO and PL cells are enriched.”

Following these leads, we modify the y-axis labelling of this graph from FDR to adjusted p-value (glmLRTTest), see Figure 6 below, and differentially expressed analysis part in the method section, page 17, “glmLRT that performs the likelihood ratio method incorporating the uncertainty in the count estimation while calculating the significance of DE detection *inbuilt in the edgeR package*”.

Figure 6

4) There is no consideration of presence/changes of a brain receptor for EPO. Presumably these data exist in the data set, and if so, would deserve comment.

Absolutely! We have responded to a similar question of Reviewer 1 and extended our discussion section in the revised manuscript, page 15.

'While our survey of neuronal lineages in the hippocampus is the broadest of its kind, it remains limited by the constraints and shortcomings of snRNA-seq. The expression level of any given gene may be underestimated due to dropout effects. In consistence with previous reports on dropout effects of this methodology, our snRNA-seq data did not or hardly detect the expression of EPOR and EPO, respectively (Ehrenreich et al 2022). Their expression, however, has been identified employing the more sensitive in situ hybridization (Wakhloo et al 2020). Moreover, RNA expression levels might not be proportional to protein abundance; thus, we ought to corroborate in future studies our observations by quantifying protein expression in situ.'

5) There are a multitude of abbreviations. A supplemental table listing all of these would be very helpful.

Thank you for making us aware! A list of abbreviations has now been added to the paper on page number 38

Table 1:

Table 1 Abbreviation of distinct pyramidal sub-lineages	
CA (1 2 3)	cornu Ammonis (1 2 3)
CA (1 3) (D)	CA1 or CA3 on the Dorsal axis
CA1 (S)	CA1 in Superficial layer
CA1 (D.F.)	CA1 in the Deep layer with Firing characteristics
Nf.M	Newly-formed and Migratory
Nf.M.Ser.	Newly-formed, Migratory, and Serotonergic
NP SUB	Near-Projecting in SUBiculum area
Nf.M.V.	Newly-formed and Migratory on the Ventral axis
Nf.M.S	Newly-formed and Migratory in Superficial layer
Nf.M.F.Ser	Newly-formed, Migratory, Firing and Serotonergic
Nf.S	Newly-formed in Superficial layer

Table 1 shows the abbreviation of pyramidal lineages classified using snRNA-seq data analysis based on their predicted location and potential functional characteristics.

There are some issues with the figures:

Fig. 1 is helpful, but would benefit from a short legend which recapitulates what is portrayed, especially the electrophysiology.

Thank you! This has now been added. Our legend reads, "Cartoon illustration of the present study design, workflow schematics and overview of our findings.

Our study commences with the 3-week rhEPO (N=6) and PL (N=6) treatments in male C57BL/6 mice. After the last injection, we processed 12 samples, 6 each from EPO and PL mice, each a pool of 2 right hippocampi (i.e. a pool of 2 mice). These samples were then subjected to snRNA-seq and the resulting data strategically analyzed to investigate the molecular, pseudo-temporal and empirical changes that occur in pyramidal lineages following EPO treatment. Finally, we performed a series of single-cell electrophysiology experiments to demonstrate that EPO affects excitatory and inhibitory input to newly formed and pre-existing pyramidal neurons in mouse hippocampi.”

Fig 2: classification was carried out by pooling both the control and treated animals. The analysis was constructed by integrating separate samples (Suppl Fig 1) but were the cellular compositions between the two groups compared? Although this work focuses on pyramidal neurons, it would be helpful to know whether there are differences in other cellular populations (e.g., neuroimmune cells, glia, etc) which may be affected to EPO exposure, as would be predicted by prior work.

Thank you! We provide a new figure on the right, which is now added to Supplementary Figure 1b. Also, we added the figure legend on the page 35 b. The stacked barplot illustrates the composition of 10 major hippocampal and a neuroimmune cell type in each EPO and PL sample based on the annotations provided in Fig. 2b

Legend of Fig 4 refers to “differentially expressed genes between pyramidal lineages shown in Fig 4b.” Should this not be Fig 3b?

Thank you again. This should indeed be Fig 3b. It is obviously a typo which we have corrected now.

Legend Fig 5 c refers to Fig. 4-5a. This is unclear to me. Is this referring to Fig 5b?

This is a typo, too; it happened during the reordering of the figures. The Reviewer is correct, It was intended to refer to the 6 lineages shown in Figure 4b. The figure legend is now corrected and reads as follows. “Heatmap representing the differential expression of genes between EPO and PL samples in the individual lineages shown in Fig. 4b. Only those genes showing an adjusted p-value <0.05 in any of the comparisons are presented here. The number of detected differentially expressed genes in each lineage is shown at the top of the heatmap.”

Fig 7b: is the difference between old/new EPO neurons significant?

Yes, it is significant. We missed adding the statistical test to the mentioned comparison. We have now added the p-values using the t-test in paired mode. See the modified Figure 7 in the manuscript, also shown below.

Suppl Fig 3: is there a key for the group numbers shown in the EPO+placebo group? Suppl Fig 7 has annotations relevant for Suppl Fig 3, but these are color coded (the colors differ between the figures) and are not enumerated. Please harmonize the figures.

We think that the Reviewer is correct. Pooling all the newly formed lineages into one was an oversight. We had pooled all newly-formed lineages into one so that the stacked barplot of composition analysis is interpretable to the audience. Following this Reviewer’s concern, we will change it back to what was shown in Supplementary Fig 3, which we have corrected in the revised version. **See Figure below).**

Extended Data Fig. 7

Suppl Fig 4: individual data points are not visualized. Do any of these regions differ significantly?

Extended Data Fig. 4

Thank you for asking. Although there is a trend that CA1 mature neurons increase upon the EPO treatment, which is also evident from the Fisher-exact test of fraction analysis, the statistical t-test does not turn out to be significant when applied to 6 (EPO) versus 6 (PL) comparisons. We have now **labelled** the figure with the individual data points and statistical test results. **See Figure on the right.** We also added the description in the figure legend as **“grey circles on the bars represent the individual data points corresponding to each sample”.**

Suppl Fig 5: One panel is unlabeled. In panel a, what are the units of the y axis?

The Y-axis is “Percentage of cell type composition”. We have added labels to the Y-axes in Supplementary Figure 5. (See Figure on the right). We also added the individual data points on the plot and description in the figure legend as “grey circles on the bars represent the individual data points corresponding to each sample”.

Suppl Fig 7b: The number of DEGs are shown on the side, not the bottom of the heatmap as stated in the legend.

Thank you for pointing it out to us. It was a mistake while trying to fit the figure in landscape mode. We have corrected the figure legend. It now matches the direction of the text in Suppl Fig 7b. We wrote on page 24: The number of detected differentially expressed genes in each lineage is shown on the right side of the heatmap. See Figure below:

Extended Data Fig. 7

Reviewer #3:

The authors present an interesting study exploring the transcriptional landscape of hippocampi of mice treated with recombinant EPO. They classify the major neuronal and non-neuronal cell types from the murine hippocampus using snRNA-seq datasets from EPO treated versus untreated mice and found several distinct pyramidal lineages. They also show that recombinant PO modulates transcriptional activity of neurogenesis and synapse-associated genes. Furthermore, they use a single-cell electrophysiology approach to demonstrate that recombinant EPO affects excitatory and inhibitory input to both newly formed and pre-existing pyramidal neurons in mouse hippocampi, with newly formed neurons receiving more excitation and less inhibition under EPO treatment than pre-existing neurons. They suggest the potential clinical benefit of EPO treatment. Overall the EPO-dependency of these effects represents a substantial step forward.

We appreciate the Reviewer's time and effort in rendering and supporting our manuscript.

This is an original and well-written study. The methods are sound and the experimental work supports the conclusions and claims, however this reviewer would like to see systematic independent validations of selected subsets of differentially regulated genes (e.g. quantitative PCR-based).

We thank the reviewer for the kind words. The genes shown in our manuscript are obtained using the most robust criteria to calculate differentially expressed genes from snRNA-seq datasets. We purposefully refrained from methods that allow for over-dispersion (the one criticism, DEGs from snRNA-seq datasets get) when calculating DEGs. To overcome this and get a reliable set of DEGs, we employed the glmQLFTest, which incorporates uncertainty in the count estimation during DEGs calculations. We show the upregulation of immediate early genes (IEGs) and other important mediators, such as Bdnf, Arc, Nr4a1, Fos, Jun, Egr1/2/3, Homer1 etc. We also show that they exist in the same regulatory network, which is activated explicitly in three newly-formed pyramidal lineages of EPO animals.

There is more than one way to independently and systematically validate the differential expressed genes (DEGs). The better way is to have a transparent and reproducible result from multiple laboratories. There are already several studies that have independently validated that these target genes are differentially expressed upon EPO treatment in the hippocampus in a similar pattern as our data shows. However, there is no report suggesting otherwise.

For instance, it was shown by qRT-PCR experiments that the expression of these genes was induced upon carbamoylated erythropoietin (CEPO), suggesting that EPO, independent of hematopoiesis, induces these genes' expression (*Tiwari et al 2019*). This was shown for Fos, Egr1, Arc, Vgf, Bdnf genes from the dentate gyrus (area of hippocampal neurogenesis). Using both qRT-PCR and Western Blot, VEGF, Bdnf, Homer1 was upregulated in the hippocampus samples where EPO was induced by hypoxia (*Li et al 2020*). Similar experiments from independent study also showed the induction of Bdnf expression by rhEPO in primary hippocampal neurons of mice (*Viviani et al 2005*). Along with Bdnf, Egr1/2/4, Arc, Nr4a3 mRNA expression in neuronal cells of hippocampus was shown to have been induced by EPO by qrt-PCR experiments (*Mengozi et al 2012*). All of the above studies are independently done from different laboratories which we think is apparantly validating the DEGs we show in newly-

formed lineages of pyramidal neurons. Therefore, we suspect that it is unlikely that performing the same experiments will yield any new information.

References:

Wakhloo, D. et al. Functional hypoxia drives neuroplasticity and neurogenesis via brain erythropoietin. *Nat Commun* 11, 1313 (2020).
 Li, G. et al. FG-4592 Improves Depressive-Like Behaviors through HIF-1-Mediated Neurogenesis and Synapse Plasticity in Rats. *Neurotherapeutics* 17, 664–675 (2020).
 Viviani, B. et al. Erythropoietin protects primary hippocampal neurons increasing the expression of brain-derived neurotrophic factor. *J Neurochem* 93, 412–21 (2005).
 Mengozzi, M. et al. Erythropoietin-induced changes in brain gene expression reveal induction of synaptic plasticity genes in experimental stroke. *Proc Natl Acad Sci U S A* 109, 9617–22 (2012).

Besides, we were a little puzzled by this comment. Several studies have compared RNAseq results to qPCR data and have found excellent correlation between these methods. However, there are obvious limitations while using qRT-PCR to validate snRNA-seq datasets. The lineages where we show differential expression of these genes are subpopulations of a subset of pyramidal lineages filtered from whole hippocampi (see Figure below, now Supplementary Fig. 9): Bar plots showing the selected neurotrophic and immediate early gene expression (normalized counts relative to transcriptome) levels in multiple pyramidal lineages. *** denote the significance of the differential expression (*, p-value < 0.05, t-test; ***, p-value < 0.001).

To perform qRT-PCR, we would need to find reliable markers and methods to sort out these cells, which is extremely challenging given the spatio-temporal

properties of these cells. Moreover, qRT-PCR could give false-positive results if the obtained cell numbers are not enough, which could be a chance in our case. Because the suggested qRT-PCR works on the bulk of cells, it is not so convenient to dissect the changes occurring at single cell levels. This gets even more challenging in our case where we are dealing with the lineages that are subpopulations of a subset of pyramidal lineages filtered from the whole hippocampi. On one hand, our snRNA-seq quantifies the uniquely mappable transcript that is assessed in a more unbiased manner, normalised and scaled against the whole transcriptome. On the other hand, qPCR experiments have their own probe bias based on what region of the cDNA is amplified. By no means is this opinion exclusively ours. A plethora of studies suggests that the modern approaches towards the RNA-seq methods and data analysis are robust enough to not always require validation by qPCR and/or similar approaches.

References:

- Coenye, T. Do results obtained with RNA-sequencing require independent verification? *Biofilm* 3, 100043 (2021).
- Everaert, C. et al. Benchmarking of RNA-sequencing analysis workflows using whole-transcriptome RT-qPCR expression data. *Sci Rep* 7, 1559 (2017).
- Wu, A. R. et al. Quantitative assessment of single-cell RNA-sequencing methods. *Nat Methods* 11, 41–6 (2014).
- Tiwari, N. K., Sathyanesan, M., Schweinle, W. & Newton, S. S. Carbamoylated erythropoietin induces a neurotrophic gene profile in neuronal cells. *Prog Neuropsychopharmacol Biol Psychiatry* 88, 132–141 (2019).
- Li, G. et al. FG-4592 Improves Depressive-Like Behaviors through HIF-1-Mediated Neurogenesis and Synapse Plasticity in Rats. *Neurotherapeutics* 17, 664–675 (2020).
- Viviani, B. et al. Erythropoietin protects primary hippocampal neurons increasing the expression of brain-derived neurotrophic factor. *J Neurochem* 93, 412–21 (2005).
- Mengozi, M. et al. Erythropoietin-induced changes in brain gene expression reveal induction of synaptic plasticity genes in experimental stroke. *Proc Natl Acad Sci U S A* 109, 9617–22 (2012).

We however, acknowledge the reviewer's concern in the manuscript by adding the following sentence on page 14: "Intriguingly, various studies have independently reported EPO-mediated induction of IEGs that we here demonstrate to be upregulated in EPO samples from newly-formed lineages. Resolving this topic further, we show that the induction of IEGs are specific to newly-formed lineages (Supplementary Fig. 9), in agreement with the potential of EPO to re-wire cognition-associated transcriptional networks."

References:

- Tiwari, N. K., Sathyanesan, M., Schweinle, W. & Newton, S. S. Carbamoylated erythropoietin induces a neurotrophic gene profile in neuronal cells. *Prog Neuropsychopharmacol Biol Psychiatry* 88, 132–141 (2019).
- Li, G. et al. FG-4592 Improves Depressive-Like Behaviors through HIF-1-Mediated Neurogenesis and Synapse Plasticity in Rats. *Neurotherapeutics* 17, 664–675 (2020).
- Viviani, B. et al. Erythropoietin protects primary hippocampal neurons increasing the expression of brain-derived neurotrophic factor. *J Neurochem* 93, 412–21 (2005).
- Mengozi, M. et al. Erythropoietin-induced changes in brain gene expression reveal induction of synaptic plasticity genes in experimental stroke. *Proc Natl Acad Sci U S A* 109, 9617–22 (2012).

REVIEWERS' COMMENTS

Reviewer #1 (Remarks to the Author):

Singh et al. carried out transcriptional hippocampal profiling of ~108,000 single nuclei and elucidated gene regulatory networks underlying fat determination of CA1 pyramidal neurons derived from multiple predecessor lineages. EPO treatment modulated neuronal function and networks and upregulated genes for neurodifferentiation, memory formation and cognition and other processes and differentially affects excitatory and inhibitory inputs.

Singh et al. addressed the concerns raised in the previous review. The text has been extended to add discussion/clarification about 1) Potential contribution of increased oxygen delivery to brain resulting from increased red blood cell production due to 3 weeks of EPO treatment; 2) Limitations of sc/snRNA-seq in determination of EPO and EPOR expression and the general problem of dropout effects; and 3) Both younger and older mice exhibit the increment of mature pyramidal neurons up to 20% and the temporary decrease in precursor cells upon EPO treatment due to differentiation returned to control levels after few weeks.

Reviewer #2 (Remarks to the Author):

My comments and concerns have been fully addressed and I have nothing additional to add. The authors are to be commended on conducting a very interesting study.

Reviewer #3 (Remarks to the Author):

The authors did an excellent job in revising their manuscript.

Point-to-Point Response to the reviewer's comments.

- Black: Reviewer's points
- Red: Our Point-to-Point Response

REVIEWERS' COMMENTS

Reviewer #1 (Remarks to the Author):

Singh et al. carried out transcriptional hippocampal profiling of ~108,000 single nuclei and elucidated gene regulatory networks underlying fat determination of CA1 pyramidal neurons derived from multiple predecessor lineages. EPO treatment modulated neuronal function and networks and upregulated genes for neurodifferentiation, memory formation and cognition and other processes and differentially affects excitatory and inhibitory inputs.

We thank the reviewer for the assessment of our findings appropriately.

Singh et al. addressed the concerns raised in the previous review. The text has been extended to add discussion/clarification about 1) Potential contribution of increased oxygen delivery to brain resulting from increased red blood cell production due to 3 weeks of EPO treatment; 2) Limitations of sc/snRNA-seq in determination of EPO and EPOR expression and the general problem of dropout effects; and 3) Both younger and older mice exhibit the increment of mature pyramidal neurons up to 20% and the temporary decrease in precursor cells upon EPO treatment due to differentiation returned to control levels after few weeks.

We thank the reviewer again for properly summarising the necessary revision we have added to this manuscript following the reviewer's suggestions.

Reviewer #2 (Remarks to the Author):

My comments and concerns have been fully addressed and I have nothing additional to add. The authors are to be commended on conducting a very interesting study.

We are grateful for the reviewer's kind words. We feel commended by your positive remarks on our study.

Reviewer #3 (Remarks to the Author):

The authors did an excellent job in revising their manuscript.

Thank you, and we extend our gratitude towards the appreciation you have conveyed to us.